

# Level up your coding: a systematic review of personalized, cognitive, and gamified learning in programming education

Kashif Ishaq[1], Atif Alvi[1], Muhammad Ikram ul Haq[2], Fadhilah Rosdi[3], Abubakar Nazeer Choudhry[4], Arslan Anjum[1] and Fawad Ali Khan[1]

[1] School of Systems and Technology, University of Management and Technology, Lahore, Pakistan
[2] Department of Computer Science, Superior University, Lahore, Pakistan
[3] Faculty of Information Science and Technology, Universiti Kebangsaan Malaysia, Bangi, Malaysia
[4] Research, Innovation & Commercialization, University of Sargodha, Sargodha, Sargodha, Pakistan

Corresponding authors
Kashif Ishaq,
kashif.ishaq@umt.edu.pk
Fadhilah Rosdi,
fadhilah.rosdi@ukm.edu.my

## ABSTRACT

Programming courses in computer science play a crucial role as they often serve as students' initial exposure to computer programming. Many university students find introductory courses overwhelming due to the vast amount of information they need to grasp. The traditional teacher-lecturer model used in university lecture halls frequently leads to low motivation and student participation. Personalized gamification, a pedagogical approach that blends gamification and personalized learning, offers a solution to this challenge. This approach integrates gaming elements and personalized learning strategies to motivate and engage students while addressing their individual learning needs and differences. A comprehensive literature review analyzes 101 studies based on research design, intervention, outcome measures, and quality assessment. The findings suggest that personalized gamification can enhance student cognition in programming courses by boosting motivation, engagement, and learning outcomes. However, the effectiveness of personalized gamification depends on various factors, including the types of gaming elements used, the level of personalization, and learner characteristics. This article offers insights into designing and implementing effective personalized gamification interventions in programming courses. The findings may inform educators and researchers in programming education about the potential benefits of personalized gamification and its implications for educational practice.

# INTRODUCTION

As per numerous institutional forecasts, computer science and related disciplines are expected to experience significant growth within the educational domain (*Venter, 2020*). However, despite this positive outlook, students often feel anxious about programming courses, finding them daunting and intimidating, which in turn hampers their motivation, engagement, and academic performance. Traditional teaching methods for computer programming, typically relying on lectures, lack the necessary interactivity, leading to

decreased student attention (*Arif, Rosyid & Pujianto, 2019*). Additionally, delivering abstract concepts through text or speech in conventional classrooms may limit students' understanding, depending on their prior programming experience (*Azmi, Iahad & Ahmad, 2015*). In response to these challenges, two prominent strategies have emerged in recent years to improve programming language education: personalization and gamification (*Ishaq & Alvi, 2023*).

Gamification involves integrating game-like elements to enhance engagement and motivation (*Ishaq et al., 2022*). This study examined the impact of implementing gamification principles, such as increasing engagement, providing learners with autonomy, and enabling progress tracking, in programming e-learning platforms on student achievement. Comparative analysis showed that participants exposed to gamified platforms had an average success rate of 84.14%, surpassing those using non-gamified systems (*Pradana et al., 2023*). Educators aim to make programming language education more enjoyable and interactive by incorporating techniques like point systems, badges, and leaderboards (*Imran, 2019*; *Knutas et al., 2014*). Gamification can particularly motivate students who are familiar with video games. While 85% of teachers recognize its benefits, technology-based gamification can enhance student motivation and learning outcomes, ultimately boosting engagement and enthusiasm in the learning process (*Rakhmanita, Kusumawardhani & Anggarini, 2023*). Building on this, personalized gamification seeks to tailor gamified elements to individual learners' preferences, learning styles, and proficiency levels. However, existing research often adopts a categorical approach, grouping students based on predetermined characteristics. This rigidity necessitates reevaluation, as it limits adaptability over the course duration. Progressive personalization, on the other hand, provides a dynamic framework that accommodates learners' evolving needs and competencies over time. This approach is advantageous, as it offers advanced students a personalized learning experience different from that of beginners. Nevertheless, the effectiveness of these strategies depends on how well they align with the cognitive processes inherent in learning and problem-solving. Therefore, gaining a comprehensive understanding of how gamification and personalization can enhance cognitive processes in programming language education is crucial (*Arkhipova et al., 2024*; *Ishaq & Alvi, 2023*). Portions of this text were previously published as part of a preprint (*Ishaq & Alvi, 2023*).

This systematic literature review provides a comprehensive overview of recent advancements in programming language education, focusing on personalization, gamification, and cognition. The review critically evaluates existing literature on techniques, frameworks, and their effectiveness in supporting cognitive processes in this field. It also identifies gaps in current research, highlighting the need for further investigation. The implications of these findings for educational practice are significant, offering valuable insights to guide the design and implementation of effective personalized gamification interventions in programming courses. Importantly, the review emphasizes the crucial role of cognition in programming language education, covering mental processes such as attention, memory, perception, reasoning, and problem-solving. A thorough understanding of these cognitive foundations is essential for educators aiming to

develop curricula and interventions that promote enhanced learning and problem-solving skills (*Ishaq & Alvi, 2023*).

The subsequent sections of this article are structured as follows: "Methodology" offers background on pertinent literature, covering topics such as personalization and gamification in education, cognitive skills, and programming language education. "Data Analysis" details our methodology for conducting the systematic literature review. "Evaluation and Deliberation on Research Questions" presents the outcomes of our review, encompassing an analysis of the primary themes and trends in the literature. "Discussion and Future Implications" explores the implications of our findings and identifies avenues for future research. Finally, "Findings, Challenges, and Recommendations" concludes the article and summarizes our key contributions.

## Background

In the evolving technological landscape, programming language education assumes a critical role in equipping students with the requisite skills for software development and coding. While gamification has garnered widespread adoption across various educational domains, its application within personalized frameworks tailored specifically for programming courses remains comparatively underexplored in extant research when juxtaposed with non-personalized counterparts. Principal methodologies in this domain entail the development of online learning platforms that customize the availability of gamification elements based on students' categorization into predetermined cohorts (*Santos et al., 2021*). However, a conspicuous void in current scholarship pertains to the imperative for greater fluidity and adaptability in personalization mechanisms; the prevailing static allocation of students into preassigned groups necessitates a paradigm shift towards dynamic adjustments predicated on real-time feedback and evolving performance metrics (*Rodrigues et al., 2021*). Moreover, gamification is using game elements in non-game contexts, often through digital platforms or applications (*Hong, Saab & Admiraal, 2024*). They focused on tailored digital gamification in education, exploring the approaches and clusters of game elements used. The performance cluster was the most commonly used, with personalized and adaptive approaches being applied in some cases. The role of gamification in students' learning and motivation is controversial, as game elements can affect individual students differently. Tailored gamification aims to improve student experiences by considering individual needs and preferences. The study identified five clusters of game elements and found ramifications for teachers who want to gamify their classes (*Ishaq & Alvi, 2023*).

The prevailing understanding suggests that while non-personalized gamification methodologies may not unequivocally enhance cognitive capabilities (*Sanmugam, Abdullah & Zaid, 2014*), they exhibit heightened efficacy under conditions eliciting negative emotional states. Moreover, it is noteworthy that research endeavors in this domain predominantly pivot to psychological paradigms rather than technological orientations (*Mullins & Sabherwal, 2018, 2020*).

## Gamification

Gamification has become a popular approach to address the issue of low motivation and engagement in e-learning platforms. Researchers have conducted various studies to explore the impact of gamification on students' motivation and engagement in education (*Ishaq & Alvi, 2023*). *Oliveira et al. (2023)* conducted a review study and discussed tailored gamification in education to combat issues like evasion, disengagement, and motivation deficit. The study found personalized gamified education based on learner traits and gamer types. The effectiveness of personalized gamification on learning outcomes remains unclear due to methodological limitations. Tailoring gamification at content and game element levels is crucial, and there is a need to compare tailored systems with non-tailored ones to enhance learning outcomes. *Shortt et al. (2023)* explored gamification in mobile-assisted language learning, focusing on Duolingo. The study indicated increased motivation and engagement in gamified environments whereas there is a need for more research on the specific aspects of gamification that impact learning outcomes. *Huseinović (2024)* found the potential drawback of using ELL games exclusively for learning English, citing potential boredom and decreased motivation. There is a lack of research on gamification in higher education in Bosnia and Herzegovina and examined how gamification impacts motivation and performance in learning English as a foreign language, focusing on variables such as proficiency, motivation, learning outcomes, and skill development. Additionally, gamification facilitates remote and distance learning, expanding access to English language education.

*Permana, Permatawati & Khoerudin (2023)* conducted a study on gamification for foreign language learning using Quizzes. The gamified nature of Quizizz contributes to a positive classroom atmosphere and fosters competition, which enhances motivation and engagement. The study examined students' perceptions of Quizzes and found it enjoyable, motivating, and engaging. Students also had a favorable view of Quizizz as a formative test tool for measuring grammatical skills and vocabulary mastery. Educators should consider individual learning styles and preferences when implementing gamification tools in the classroom. *Dehghanzadeh et al. (2024)* highlighted the growing interest in gamification in K-12 education, citing its potential to enhance learning outcomes through increased motivation. Despite mixed results, gamification has shown promise in promoting cognitive, affective, and behavioral learning outcomes. Scholars emphasize the importance of game elements and instructional support in designing effective gamified learning environments, calling for further research to address existing gaps in the literature. *Zhang & Hasim (2023)* reviewed recent research on gamified English language instruction, noting its benefits and drawbacks. The study highlights widespread adoption in non-English-speaking countries, emphasizing the importance of designing gamified environments with attention to dynamics and mechanics. Positive outcomes are reported for learners of all ages and genders, underscoring the need to explore diverse game-based learning applications. The review concludes by stressing the importance of aligning gamification activities with students' educational levels, cognition, and capabilities. Portions of this text were previously published as part of a preprint

(*Durst & Henschel, 2024*). *Venter (2020)* conducted a systematic literature review of gamification in higher education programming courses. Points, achievements, levels, leaderboards, and badges were the most commonly used gamification elements in the reviewed studies. The studies reviewed by Venter showed that gamification positively affected engagement, motivation, and learning outcomes. Gamification in a programming language using game design elements can increase learner engagement, motivation, and retention, improving performance as learners spend more time studying and earning badges (*Imran, 2022*; *Deterding et al., 2010*). Portions of this text were previously published as part of a preprint (*Ishaq & Alvi, 2023*).

Similarly, *Ghosh & Pramanik (2023)* explored that gamification in education utilizes game elements and design principles to enhance learning outcomes and student engagement. The study found its effectiveness in computer science courses, improving motivation, knowledge retention, and understanding of complex concepts. Educators can create dynamic learning environments through gamification, fostering collaboration, competition, and active participation. Furthermore, gamified learning positively impacts student experiences and material retention. *Kiraly & Balla (2020)* developed a learning management system containing online programming language courses and added points, incentives, badges, immediate feedback, and a leaderboard to gamify the courses. Their results showed that the students who completed a Java course with gamification were better at solving coding tasks. Furthermore, *Katan & Anstead (2020)* developed a gamification platform called "Sleuth" that teaches introductory programming. They found that students who used Sleuth received a very high median grade (90.67%), while students who used a module-based testing environment received quite a relatively low grade (66.94%). Their research filled a gap in the literature by creating a gamification platform that resembles a video game. *Pankiewicz (2020)* found that gamification elements such as points, badges, and leaderboards positively impacted students' motivation in the learning process (*Ishaq & Alvi, 2023*).

*Cao (2023)* also addressed challenges for Chinese international students in programming courses and proposed story-based and AI-enhanced gamification to improve their learning experiences. The authors found positive impacts of story-based gamification on students' sense of belonging and motivation. The proposed learning system included instructional content, gamification design, user interface, and a generative language model. *Queirós (2019)* presented a framework called "PROud" that applies gamification features based on the usage data of programming exercises, such as fostering competition between students based on the correctness of code solutions submitted. Moreover, *Hassan et al. (2021)* investigated why students lack motivation in e-learning platforms and concluded that it stems from their learning experience. The studies suggest that gamification elements in e-learning platforms can significantly improve students' motivation and engagement in programming courses. The following gamified elements are used to design games in different educational environments:

*Goal orientation:* Goal orientation in gamification involves designing educational gamified environments around distinct learning objectives, each segmented into smaller

tasks. As students master each concept, they progress through levels of increasing difficulty. This approach offers a structured and progressive learning experience within a gamified setting (*Ahmad et al., 2020*).

*Challenges:* Challenges in gamification include designing effective and engaging tasks, creating meaningful rewards, and ensuring that the gamified system aligns with the learning objectives. Additionally, it can be difficult to sustain motivation and interest over time and balance gamification's competitive aspects with collaboration and teamwork (*Suresh Babu & Dhakshina Moorthy, 2024*).

*Achievements:* Achievement in gamification is frequently associated with attaining specific goals and milestones within the gamified environment. This can encompass earning badges, unlocking levels, or completing designated tasks. By setting clear objectives and offering rewards for achieving them, gamification motivates users to strive for and celebrate their accomplishments. This approach effectively enhances motivation and engagement across various activities and learning experiences (*Ishaq et al., 2022*).

*Levels:* Levels denote different stages of progression within the gamified system. As users complete tasks, they advance to higher levels, each often presenting increased challenges or complexity. Levels can provide a sense of achievement and progression, encouraging continued engagement with the gamified experience (*Suresh Babu & Dhakshina Moorthy, 2024*).

*Progress bar:* Progress bars are frequently employed to visually depict a player's advancement toward a goal or achievement. This powerful motivational tool effectively encourages continued engagement and goal completion (*Suresh Babu & Dhakshina Moorthy, 2024*).

*Badges:* Badges are a common element of gamification, used to signify a user's accomplishments or milestones within a system that visually represents progress and achievement. They can be awarded for various actions or goals, motivating users to engage more with the platform and strive to earn additional badges. Achievement systems often inspire the concept of badges in gamification in video games, where players are awarded badges or trophies for completing specific tasks or reaching particular milestones (*Zourmpakis, Kalogiannakis & Papadakis, 2023*).

*Leaderboard:* A leaderboard in gamification ranks and displays participants' performance in a game or activity. It showcases players' scores, achievements, or progress in a competitive or collaborative environment, facilitating comparison and competition. Leaderboards are widely used to promote engagement and healthy competition among users, fostering a sense of accomplishment and motivating individuals to improve their performance to climb the ranks. This gamification element is commonly employed in diverse contexts, including educational platforms, fitness apps, and employee training programs (*Ishaq et al., 2022*; *Zourmpakis, Kalogiannakis & Papadakis, 2023*).

*Reward:* Rewards are incentives provided to users for completing tasks, reaching milestones, or demonstrating desired behaviors within a game, app, or system. These rewards can take various forms, including points, virtual goods, badges, discounts, or real-world items. By offering rewards, gamified systems aim to motivate and engage users, promoting continued participation and progress. This concept is rooted in behavioral

psychology, as rewards can reinforce positive actions and encourage further engagement with the gamified experience (*Ishaq et al., 2022*; *Wang et al., 2024*; *Zourmpakis, Kalogiannakis & Papadakis, 2023*).

## Personalization and adaptation

In recent years, researchers have emphasized the significance of personalization in gamification research. It has been suggested that considering users' characteristics can enhance the potential benefits of gamification (*Rajanen & Rajanen, 2017*; *Ghaban & Hendley, 2019*). Portions of this text were previously published as part of a preprint (*Ishaq & Alvi, 2023*). *Rakhmanita, Kusumawardhani & Anggarini (2023)* reported gamification's ability to motivate students familiar with video games. While 85% of teachers understand its benefits and can adapt it in their courses, some encounter challenges due to time constraints and subject adaptation difficulties. Similarly, *Zourmpakis, Kalogiannakis & Papadakis (2023)* explored adaptive gamification in science education, aiming to tailor game elements to individual user preferences. They also investigated students' motivation and engagement with adaptive gamified applications in science education, revealing strong student interest, particularly in specific game elements. Personalization can increase students' motivation and engagement by providing a tailored learning experience that caters to their needs, preferences, and performance. *Clarice et al. (2023)* explored the effectiveness of gamification on learners' academic performance. Phenomenological research was employed to understand individuals' experiences with gamification, considering factors like motivation and preparedness. Drawbacks included time commitment for resource gathering, while rewards significantly boosted engagement and performance. *Suresh Babu & Dhakshina Moorthy (2024)* explored mapping user profile elements with gamification elements using AI techniques. AI can play a crucial role in adapting gamification to user profile elements like engagement level and learner type. AI algorithms integrated into gamification frameworks can significantly improve student engagement. Adaptive learning, personalized feedback, and customization of game elements based on students' profiles are some of the gamification techniques that can be used to achieve personalization. *Rodrigues et al. (2023)* discussed that multidimensional personalization enhances students' autonomous motivation in virtual learning environments (VLE) and the advantages of gamification within them. Decision trees identify game elements based on user preferences and demographics.

*Challco et al. (2015)* studied using ontologies to personalize gamification in collaborative learning environments. They proposed that collaborative gamification techniques could address the issue of decreased motivation. Similarly, *González et al. (2016)* explored enhancing student engagement in learning systems through the personalization of gamification. They developed an intelligent tutorial system that incorporated adaptation and personalization of gamified elements. *Gharbaoui, Mansouri & Poirier (2023)* focused on enhancing engagement, motivation, and success in teaching and learning through personalized gamification and social learning. The research proposes a model integrating personalized gamification, social learning, and adaptivity to boost learner satisfaction and success rates, emphasizing customization and motivational game elements.

*Janson et al. (2023)* examined a special issue editorial on the widespread use of games and game-like elements in information systems. The editorial emphasizes adaptive and intelligent gamification designs, stressing the significance of personalized approaches. The use of gameful experiences and personalized systems is linked to increased engagement levels. Furthermore, the authors reported the potential of integrating gamification with virtual lab teaching techniques to enhance student learning outcomes. *Roosta, Taghiyareh & Mosharraf (2016)* focused on personalizing gamified elements in an online learning environment based on learners' motivation. Their study proposed characterizing various game elements and students' motivation types to create a personalized learning management system. The personalization system adapted the gamified elements displayed to students based on their motivation category. *Knutas et al. (2017)* designed a profile-based algorithm for personalization in online collaborative learning environments based on intrinsic skill atoms and gamification-based user-type heuristics. They also developed personalized gamification software using this profile-based algorithm.

Several studies have explored the benefits of personalized gamification on student engagement, motivation, and cognition. *Knutas et al. (2019)* developed a machine learning-based personalized content system, whereas *Rodrigues et al. (2022)* investigated the relevance of personalization characteristics and collected user feedback on game elements such as points and rewards. *Bennani, Maalel & Ghezala (2020)* created an adaptive gamification ontology called "AGE-Learn" and found that personalized gamification improved online student engagement, motivation, and cognition. *Santos et al. (2021)* grouped users into six categories and studied the association between user types and their feedback on different gamification elements. Personalization is particularly beneficial because students have unique learning styles, personalities, values, and motivating factors. Overall, personalization is a key aspect of gamification research, and several studies have explored how it can be used to enhance the potential benefits of gamification for learners.

**Adaptive learning technologies:**

The study of *Essa, Celik & Human-Hendricks (2023)* explored the use of artificial intelligence (AI) and machine learning (ML) techniques in personalized adaptive learning systems to identify learners' learning styles (LSs) and enhance e-learning experiences. The Felder-Silverman Learning Style Assessment (FSLSM) is commonly used in technology-enhanced learning. Technology plays a crucial role in education by revolutionizing and enhancing the learning process through adaptive learning. It offers easy access to information, personalized learning experiences, and opportunities for collaboration. AI-powered platforms analyze student data to customize learning experiences and provide early intervention for emotional distress, tailoring instruction to individual needs, preferences, and learning styles (*Aggarwal, 2023*).

**AI-driven personalized learning:**

The study of *Msekelwa (2023)* found that AI can enhance learning outcomes, increase efficiency, and offer personalized support to learners. Different groups emphasized aspects of digital learning: machine learning simplifies learning, chatbots promote critical thinking, and AI-driven language tools remove barriers. The integration of AI in education

has revolutionized traditional teaching by offering personalized learning and boosting student engagement. AI's impact extends beyond the classroom, influencing curriculum development and assessment. However, ethical challenges like privacy concerns and algorithmic biases need to be addressed. Collaboration among educators, policymakers, and technologists is essential to establish ethical guidelines and ensure equitable access to AI-enhanced educational resources (*Ayeni et al., 2024*). The study of *Rane, Choudhary & Rane (2023)*, discussed Education 4.0 and 5.0, highlighting the role of AI in Education 5.0 to provide personalized learning, enhance engagement, and deepen understanding of complex subjects. AI-driven adaptive learning systems adjust content dynamically based on individual progress, leading to more effective learning experiences. Educational chatbots can significantly enhance personalized learning by promoting self-regulated learning (SRL) through classroom AI integration. They facilitate goal setting, self-assessment, and personalization, supporting student self-regulation and providing personalized feedback. The article highlights that AI chatbots can improve academic performance and stresses the importance of incorporating pedagogical principles in their design to support student learning effectively (*Chang et al., 2023*).

**Dynamic assessments:**

The study by *Rodrigues et al. (2023)* compared multidimensional personalization to One Size Fits All (OSFA) across three institutions, involving 58 students in a controlled experiment. It examined gamification designs customized to learning tasks, users' gaming preferences, and demographics. Findings showed no significant differences in motivating students to complete learning assessments between OSFA and personalized designs, though motivation varied less with personalization. Exploratory analysis indicated that personalization benefited females and those with technical degrees but had drawbacks for individuals who prefer adventure games or solo play. *Vashishth et al. (2024)* reported that AI-driven learning analytics have the potential to offer personalized feedback and assessment to enhance student engagement and optimize educational outcomes. Ethical considerations and challenges are crucial in this evolving field. AI's integration in higher education has introduced terms such as key performance indicators (KPIs), Internet of Things (IoT), learning management systems (LMS), Artificial Intelligence (AI), and machine learning (ML).

**Personalized learning environments & content delivery:**

The study by *Ismail et al. (2023)* discussed the typical architectural features of personalized learning software, focusing on aspects that support the software's functionality. They analyzed 72 systems, proposed a taxonomy, and identified three main architectural components: the learning environment, learner model, and content. The study primarily focuses on formal software systems and provides guidelines for researchers and practitioners. The study by *Zhong (2023)* examined the design elements of personalized learning, concentrating on three critical areas: structuring learning content, sequencing learning materials, and supporting learning readiness. *Rodrigues et al. (2023)*, emphasize the critical role of virtual learning environments in education, mainly through personalized gamification. Their study highlights that personalization ensures equitable experiences across user groups, rather than just increasing average outcomes. They stress

the importance of considering factors like gender, education level, preferred game genre, and playing environment when implementing personalized strategies.

## Cognitive skills and learning process

**Microlearning:**

The DIL-MicLearn system is a personalized online learning platform that combines mastery learning, microlearning, and adaptive learning. It aims to provide personalized learning experiences, improve learning outcomes, and boost student satisfaction. The system features small units of learning content, direct feedback, and regulatory control for teachers, which help reduce cognitive load and enhance student satisfaction (*Marti et al., 2024*).

Research has shown that gamification can positively affect cognitive processes, such as attention, memory, and learning (*Hamari, Koivisto & Sarsa, 2014*). *Scamardella, Saraiello & Tafuri (2023)* reported that "Learning through play" facilitates the development of life skills and the learning process of educators, fostering social interaction, cognitive growth, emotional maturity, and self-assurance crucial for tackling life's hurdles. *Amer et al. (2023)* discussed the effectiveness of Sokoon, a gamified cognitive-behavioral therapy (CBT) app, in alleviating depression, anxiety, and stress among university students and teenagers. The app offers evidence-based CBT skills and customizable features, significantly improving depression, sleep quality, and quality of life. The study evaluates the implementation of gamified CBT for depressive and associated symptoms, incorporating techniques like Hexad theory and dynamic difficulty adjustment (DDA) to personalize interventions.

Gamification can enhance cognitive engagement by encouraging active information processing, focused attention, and decision-making based on feedback provided by the game. Several studies have found that gamification can improve cognitive engagement and learning outcomes. For instance, *Rojas-López et al. (2019)* explored gamification's impact on engagement in higher education programming courses. Their study emphasized gamification's emotional and social aspects, stating that recognizing students for their accomplishments through awards, trophies, or achievements can provide emotional motivation, and encouraging students to work together to complete a task can provide social motivation. The results indicated that gamification significantly improved student engagement. Portions of this text were previously published as part of a preprint (*Ishaq & Alvi, 2023*). *Clarice et al. (2023)* explored the effectiveness of gamification on learners' academic performance and its ability to motivate and engage through features like scoring and competition. Learning theory informed meaningful gamification implementation, with research indicating higher academic performance in gamified environments. Additionally, the study also examined the impact of gamification on elementary teachers and students, emphasizing the importance of student participation and performance for success.

Furthermore, *Erlangga et al. (2024)* explored that gamification was applied within the Learning Management System to enhance student motivation and cognition. Using the ADDIE method, creating a Smart Learning Environment and analyzing pre-test and post-test results using N-Gain calculation showed a significant cognitive improvement in

the gamification group (N-Gain score: 26.6111), compared to the non-LMS group (N-Gain score: −19.8889). *Mullins & Sabherwal (2020)* approached gamification from a cognitive-emotional perspective. They highlighted the importance of considering both positive and negative emotions in gamification. They suggested that emotions and cognitions can interact further to enhance the positive outcomes of a gamified system. Recently, there has been growing interest in exploring the potential of combining these three areas to improve the effectiveness of programming language education. Educators hope to increase student engagement and motivation by personalizing instruction and incorporating gamification elements, promoting more effective learning outcomes. Additionally, understanding the role of cognition in programming language education is crucial for designing effective curricula and interventions. Educators can design interventions that promote more effective learning and problem-solving in programming language education by understanding how these processes work and how they can be supported. Portions of this text were previously published as part of a preprint (*Ishaq & Alvi, 2023*).

**Conceptual understanding:**

In conceptual understanding a study by *Hardiansyah et al. (2024)*, assessed the Science Problem Solving Test, explored cognitive styles, and found that field-independent students excelled in analytical problem-solving while field-dependent students faced difficulties. The study concluded that cognitive style significantly impacts scientific attitudes and knowledge competence. *Hurtado-Bermúdez & Romero-Abrio (2023)* in a Forensic Physics course assessed the impact of combining virtual and research labs on learning about electron microscopes. Using both lab types significantly improved students' understanding of complex concepts but did not increase interest in scientific careers. Another study by *Kong, Cheung & Zhang (2023)* evaluated an AI literacy program for 36 university students from various disciplines, focusing on conceptual understanding, literacy, empowerment, and ethical awareness. The program included 7 h on machine learning, 9 h on deep learning, and 14 h on application development. Assessments showed significant improvements in AI knowledge and ethical awareness. A study on tenth graders' understanding of force and motion found that metacognitive instruction was more effective than traditional methods. Students with higher pre-epistemic cognition gained the most, highlighting the value of metacognitive strategies in enhancing conceptual understanding (*Yerdelen-Damar & Eryılmaz, 2021*).

**Cognitive load**

A study by *Chen et al. (2023)* determined that task complexity, driven by element interactivity, is crucial in human performance and behavior. It noted that complexity is affected by information structure and long-term memory. The review underscores the importance of considering element interactivity in instructional design to enhance learning and reduce cognitive load. The study of *Zhang et al. (2023)* explored how teaching presence impacts students' emotional engagement through the cognitive load, with the moderating effect of the need for cognition. A survey of 883 university students found that teaching presence enhances emotional engagement by influencing cognitive load. Higher levels of need for cognition amplified teaching presence's positive impact and mitigated

cognitive load's negative effect on emotional engagement. These findings advance our understanding of how instructional factors shape students' motivational outcomes, aligning with expectancy-value and cognitive load theories. The study of *Bahari (2023)* investigated cognitive load management in technology-assisted language learning (TALL) environments. It identified eighteen tools, such as visualization aids and dual computer displays, to help teachers and learners manage cognitive load. The study also outlined seven challenges, including adjusting task difficulty and adapting design principles, which are crucial for future TALL research.

**Misconceptions:**

A study by *Font, Burghardt & Leal (2023)* reported that reptiles' cognitive abilities have often been underestimated due to misconceptions about their brain structure. Recent research reveals that reptiles possess complex brain structures akin to mammals and birds, enabling behaviors like spatial learning, social learning, problem-solving, and communication. The study reported that cognition and learning are continuous processes shaped by interactions within dynamic systems. It also explored how variation, fluctuation, and context influence students' thinking and learning. A dynamic systems perspective views misconceptions not as fixed entities but as patterns emerging from complex systems. This perspective encourages valuing all conceptions and understanding the fluidity of students' thinking (*Gouvea, 2023*). The study of *Lagoudakis et al. (2023)* investigated the relationship between hemispheric preference and students' misconceptions in biology, finding no significant difference in the number of misconceptions between those with right-hemisphere dominance and left-hemisphere dominance. Conducted with 100 seventh-grade students using a correlational explanatory approach, it revealed that 60% were left-brain dominant, 36% were right-brain dominant, and 4% were whole-brain dominant. Similarly, *Berweger, Kracke & Dietrich (2023)* examined how discovering confidently held misconceptions influences emotions and motivation among 275 preservice teachers assessing statements about education. Feedback based on scientific evidence revealed that participants felt more surprise, curiosity, confusion, and anger when high-confidence misconceptions were disproved compared to low-confidence ones.

**Metacognition**

*Tucel Deprem et al. (2023)* found that argument-based inquiry (ABI) instruction was more effective than traditional lectures and structured activities. The ABI group achieved higher science content understanding, better metacognition, and more developed epistemological beliefs. A study by *Yerdelen-Damar & Eryılmaz (2021)* on metacognitive instruction for tenth graders found it more effective than traditional teaching in understanding force and motion. Students with higher pre-epistemic cognition benefited more, underscoring the importance of metacognitive strategies for deeper conceptual learning. The impact of metacognitive interventions on knowledge transfer among students was assessed, and it was found that nudges benefited factual learners and practice-aided procedural learners, helping both groups match the performance of conditional

learners on logic and probability tutors. The results suggested that these interventions effectively facilitated knowledge transfer (*Abdelshiheed et al., 2024*). The study by *Ulu & Yerdelen-Damar (2024)* examined physics identity and found that gender differences in physics self-efficacy could explain variations in physics identity, recognition, and interest. It also found that metacognition and epistemic cognition indirectly influenced physics identity through their impact on physics self-efficacy. Previous research has explored various aspects of personalization, gamification, and programming language education and has highlighted the potential benefits of each (*Ishaq & Alvi, 2023*). However, no systematic literature review examines the state-of-the-art in personalization, gamification, cognition, and programming language education and how these areas intersect. Our current article aims to fill this gap by providing a thorough overview of the existing literature and identifying areas for further research.

## METHODOLOGY

This section outlines the systematic literature review process to identify and analyze relevant studies on personalized gamification, cognition, and programming language education. It describes selecting appropriate search terms, databases, and inclusion and exclusion criteria. Furthermore, it outlines the screening and selection procedure, as well as the techniques employed for data extraction and quality assessment to ensure the dependability and accuracy of the results. The following subsection outlines the research questions. Portions of this text were previously published as part of a preprint (*Ishaq & Alvi, 2023*).

### Research questions

**RQ1:** What dataset is available to other researchers to establish an article library, and what are the trends, publication channels, and geographical areas in personalized gamified programming education?

**RQ2:** What criteria can be used to assess the quality of articles selected for review in the context of personalized gamified programming education, and how can these criteria be applied to the articles identified in RQ2 to ensure that only high-quality research is included in the dataset?

**RQ3:** What are the prevailing trends and optimal methodologies for integrating personalized gamification frameworks in programming education, and what distinctions exist in the design and customization of these frameworks?

**RQ4:** How might personalized gamification frameworks in programming education be correlated with the various cognitive domains delineated within Bloom's taxonomy?

**RQ5.** What tools and software applications are developed based on personalized gamification frameworks in programming education, and how are these tools tailored to specific programming languages and concepts?

**RQ6:** What are the common processes, tools, and instruments utilized to evaluate applications based on personalized gamified programming education? What evaluation measures are employed to assess applications from various viewpoints, such as teaching, learning, and technical perspectives?

### Research design
#### *Digital library and search strategy*

The search strategy for this study was designed to identify relevant articles based on the research questions. The specific details of the search strategy are presented in the following subsections.

#### *Automated search in Web of Science (WoS core collection)*

A systematic investigation was carried out to filter irrelevant research and obtain adequate information. The Web of Science Core Library is a curated database of over 21,100 peer-reviewed journals, including top-tier academic journals worldwide (including Open Access journals), covering over 250 disciplines (Universities). It is widely regarded as a tool that helps users efficiently gather, analyze, and share information from various databases (*Clarivate, 2023*). To conduct the systematic literature review (SLR) in an organized and efficient manner, the researcher used this platform to retrieve research articles by combining 'AND' and 'OR' Boolean operators with keywords to create a search string. Figure 1 provides an overview of the search results obtained from the Web of Science. Table 1 presents the ultimate search string, which utilized 'AND' and 'OR' Boolean operators with keywords to query the WoS Core Collection. The search was limited to titles only, and a filter based on indices and time span was applied to narrow down the search query for the study.

#### *Inclusion criteria*

The article included in the review must be in the domain of personalized, cognition, and game-based computer programming learning that must target the research questions. The article published in journals or conferences from 2014 through 2024 is included in the review.

#### *Exclusion criteria*

Articles excluded from the study that were not written in English were not accessible, and also that do not discuss or focus on personalized, cognition, and game-based computer programming in educational institutes. A detailed flowchart of inclusion/exclusion criteria is presented in Fig. 1.

#### *Skim and scan screening*

The screening process consisted of two stages: title and abstract screening and full-text screening. Two reviewers independently screened the titles and abstracts of all identified articles against the inclusion criteria. After title and abstract screening, two independent reviewers retrieved and reviewed full-text articles against the inclusion criteria. Any discrepancies were resolved through discussion between the two reviewers. The screening process followed the PRISMA guidelines (*Ishaq et al., 2021*) and is presented in a flow diagram in Fig. 1.

### Data extraction

Two independent reviewers conducted the data extraction process following the PRISMA guidelines. The reviewers utilized a pre-designed data extraction Excel sheet to collect
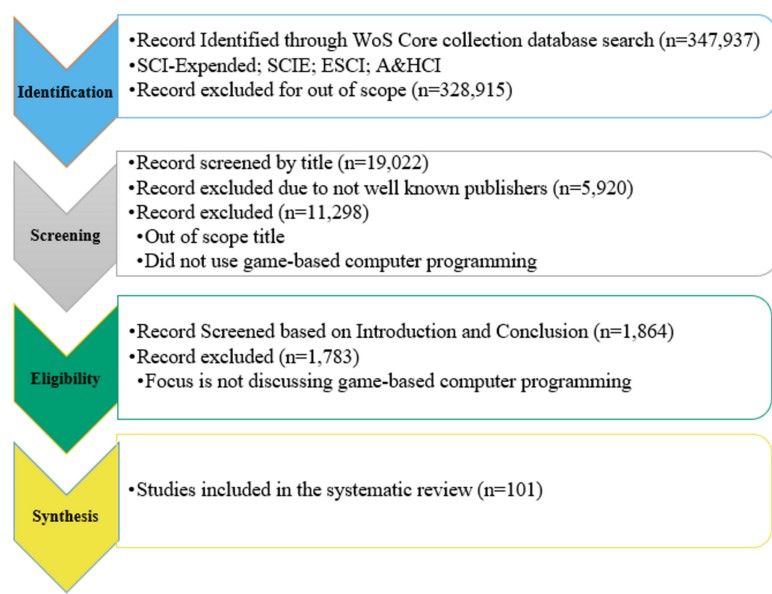

**Figure 1 Flow chart of systematic review process.**

**Table 1 Search strategy for digital library.**

| Digital library | Search query | Applied filter |
|---|---|---|
| (WoS Core Collection)<br><br>SCI-Expanded<br><br>SCIE<br><br>ESCI<br><br>AHCI | Gamification (Title) OR gamified (Title) OR game (Title) OR game-based (Title) OR game-based (Title) OR serious game (Title) AND programming (Title) OR programming (Title) OR programming course (Title) OR programming subject (Title) AND Cognition (Title) OR Cognitive Skill (Title) OR comprehension (Title) OR perception (Title) OR understanding (Title) OR learning (Title) AND personalization (Title) OR personalized (Title) OR realization (Title) OR actualization (Title) AND Adaptive (Title) OR Adaptation (Title) | 2015–2024 |

relevant information from the selected articles. The data extraction form included the following information:

- Study characteristics: authors, year of publication, title, journal/conference, country, research design, sample size, and study duration.
- Gamification and personalization features: gamification elements used, personalization techniques applied, and their effects on learning outcomes.
- Cognitive aspects: the impact of gamified and personalized programming education on cognitive skills, such as problem-solving, critical thinking, creativity, and motivation.
- Programming languages: the programming languages and concepts used in the studies.
- Evaluation methods: the evaluation methods used to measure the effectiveness of gamified and personalized programming education.

**Table 2 Possible rating for recognized and stable publication source.**

| Sr. No. | Publication source | 4 | 3 | 2 | 1 | 0 |
|---------|-------------------|-----|-----|--------|--------|---------------------|
| 1 | Journals | Q1 | Q2 | Q3 | Q4 | No JCR ranking |
| 2 | Conferences | Core A | Core A | Core B | Core C | Not in core ranking |

### Selection based on quality assessment

The collection of appropriate studies based on quality assessment (QA) is considered the key step for any review. As the fundamental studies differ in nature, the critical assessment tools (*Fernandez, Insfran & Abrahão, 2011*) and *Ouhbi et al. (2015)* used to conduct QA are also supplemented in our analysis by quantitative, qualitative, and mixed approaches. To enhance the rigor of our study, we developed a QA (quality assurance) questionnaire to assess the accuracy of the selected records. The authors conducted the QA for our research using the following parameters for each study:

1) If the analysis leads to personalized, cognition, and game-based computer programming language learning, the result is (1); otherwise, (0).
2) If the studies provide suitable methodology, then award (1) or else score (0).
3) As simple answers in results are given for personalized, cognition, and game-based computer programming language learning, the analysis will provide the following scores: 'Yes (2),' 'Limited (1),' and 'No (0).'
4) Studies have been analyzed concerning graded rankings of journals and conferences in computer science (*Ishaq et al., 2021*). Table 2 indicates potential findings for publications from known and reliable sources.

After combining the number of the above-mentioned parameters, a final score (value between 0 and 8) was determined for each study. Articles with four or more ratings were included in the final results.

### Selection based on snowballing

After conducting a standard appraisal, we utilized backward snowballing through the reference lists of any completed analyses to identify additional relevant articles (*Mehmood et al., 2020*). Only those candidate articles that satisfied the inclusion/exclusion criteria were considered. The inclusion/exclusion of an article was determined after reviewing its introduction and other relevant sections.

## DATA ANALYSIS

In this section, the overview of finalized studies is provided. Portions of this text were previously published as part of a preprint (*Ishaq & Alvi, 2023*).

### Overview of intermediate selection process outcome

Game-based programming language learning is a very active topic, and the analysis approach of the researchers is to find suitable research systematically and empirically from the Web of Science core collection. The next step after finding the relevant research is to

**Table 3 Selection phases and results.**

| Phase | Selection | Selection criteria | Indexes: SCI-EXPANDED, SSCI, A&HCI, ESCI |
|---|---|---|---|
| 1 | Search | Keywords (Figure) | 347,957 |
| 2 | Filtering | Title | 19,042 |
| 3 | Filtering | Abstract | 11,298 |
| 4 | Filtering | Introduction and conclusion | 1,864 |
| **5** | **Inspection** | **Full article** | **101** |

compile the records to form the foundation for analysis. More than 300,000 articles were found in the Web of Science core collection by providing the keywords from 2015 to 2023. Inclusion and exclusion criteria were defined for filtering the record based on titles, the abstract, articles written in English, accessibility of the document, and considering well-known publishers. Moreover, articles focused on personalized, cognition, and game-based computer programming languages in educational institutes were included in this research, whereas the non-availability of any area in the article was excluded.

### Overview of selected studies

Table 3 presents significant results of primary search, filtering, and review processes that include Web of Science indices. At the filtering/inspection stage, the automatic search decreased this amount to 101 articles.

## EVALUATION AND DELIBERATION ON RESEARCH QUESTIONS

This section analyzed 101 primary studies based on our research questions. Portions of this text were previously published as part of a preprint (*Ishaq & Alvi, 2023*). The following section presents the findings of the SLR on personalized gamified programming education:

### RQ1. What dataset is available to other researchers to establish an article library, and what are the trends, publication channels, and geographical areas in personalized gamified programming education?

Table 4 and Fig. 2 present the geographical distribution of selected studies. Most studies were from Europe, 36, whereas American countries published 24. Asian countries published 18 studies, while only three were published by the ocean and the African continent.

The data presented in Table 5 reveals that the maximum number of studies has been selected from highly recognized journals indexed in the Web of Science, and the rest of the studies picked good-ranking conferences. Education and Information Technologies is at the top of the list, with six studies selected, followed by the Interactive Learning Environment journal, with three selected. Similarly, Computer & Education, Journal of Educational Computing Research, and MDPI-Information are the journals from which three studies were selected.

**Table 4 Identified publications geographically.**

| Sr. No. | Sub-continent | Countries | Number of publication |
|---------|---------------|-----------|-----------------------|
| 1 | Europe | Greece | 9 |
| | | Spain | 6 |
| | | Portugal | 5 |
| | | Germany | 3 |
| | | Netherland | 3 |
| | | Finland | 3 |
| | | Belgium | 1 |
| | | Croatia | 1 |
| | | Estonia | 1 |
| | | Hungary | 1 |
| | | Lithuania | 1 |
| | | Mexico | 1 |
| | | Slovakia | 1 |
| | | Slovenia | 1 |
| | | Sweden | 2 |
| | | Italy | 1 |
| | | Bosnia & Herzegovina | 1 |
| | | UK | 2 |
| 2 | Asia | Malaysia | 5 |
| | | Turkey | 3 |
| | | Oman | 2 |
| | | Pakistan | 2 |
| | | China | 1 |
| | | India | 3 |
| | | Iran | 2 |
| | | Japan | 1 |
| | | Korea | 1 |
| | | Indonesia | 3 |
| | | Taiwan | 1 |
| | | Morocco | 1 |
| | | Russia | 1 |
| | | Egypt | 1 |
| | | Thailand | 1 |
| 3 | America | US | 17 |
| | | Brazil | 7 |
| | | Mexico | 2 |
| | | New York | 1 |
| 4 | Oceania | Australia | 2 |
| 5 | Africa | Tunisia | 1 |
| **Total** | | | **101** |

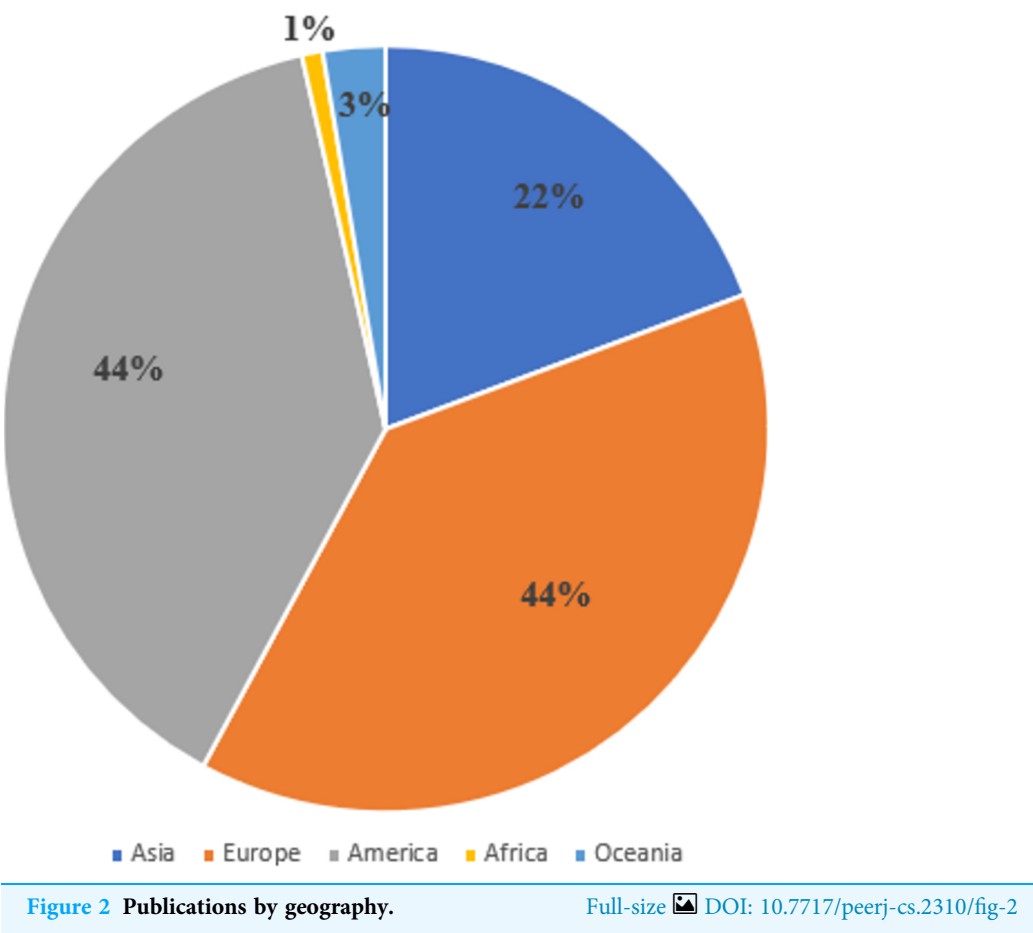

**Figure 2** **Publications by geography.**

### RQ2: What criteria can be used to assess the quality of articles selected for review in the context of personalized gamified programming education, and how can these criteria be applied to the articles identified in RQ2 to ensure that only high-quality research is included in the dataset?

The quality assessment (QA) score for each finalized study is awarded according to the criteria defined in "Data Analysis", as shown in Table 2. Further, it shows the QA score ranges from 4–8, whereas a score less than four for the studies is discarded. Game-based programming language learning researchers may find this QA helpful in choosing related studies while addressing its usage and challenges. Articles published in Q1 journals mostly scored the highest, while studies scoring four are from less recognized journals but relevant to the subject matter. 21 out of 101 scored highest, *i.e.* eight, which showed that the studies met all QA criteria, whereas 13 got the second highest score in the QA. Likewise, 12 out of 101 studies got the lowest score in the QA because they did not meet all the criteria. The overall classification results and QA of the finalized studies are presented in Table 6. Finalized studies have been classified based on five factors: empirical type, research type, and methodology.

**Table 5 Publication sources.**

| Sr. No. | Publication source | Channel | No. of articles |
|---|---|---|---|
| 1 | Education and Information Technologies | Journal | 7 |
| 2 | Interactive Learning Environments | Journal | 3 |
| 3 | Computers & Education | Journal | 4 |
| 4 | Journal of Educational Computing Research | Journal | 3 |
| 5 | MDPI-Information | Journal | 3 |
| 6 | Computer Application Engineering Education | Journal | 3 |
| 7 | Educational Technology Research and Development | Journal | 2 |
| 8 | IEEE Transactions on Learning Technologies | Journal | 2 |
| 9 | Multimedia Tools and Applications | Journal | 2 |
| 10 | Simulation & Gaming | Journal | 2 |
| 11 | ACM Transactions on Computing Education | Journal | 1 |
| 12 | Acta Didactica Napocensia | Journal | 1 |
| 13 | ARPN Journal of Engineering and Applied Sciences | Journal | 1 |
| 14 | Education Sciences | Journal | 1 |
| 15 | EMERALD INSIGHT | Journal | 1 |
| 16 | Entertainment Computing | Journal | 1 |
| 17 | Higher Education | Journal | 1 |
| 18 | IEEE Access | Journal | 1 |
| 19 | IEEE Latin America Transactions | Journal | 1 |
| 20 | IEEE-RITA | Journal | 1 |
| 21 | Informatics in Education | Journal | 1 |
| 22 | International Journal of Engineering Education | Journal | 1 |
| 23 | International Journal of Information Management | Journal | 1 |
| 24 | International Journal of Serious Games | Journal | 1 |
| 25 | International Journal of Technology Enhanced Learning | Journal | 1 |
| 26 | International Journal of Web Information Systems | Journal | 1 |
| 27 | Journal of Business Research | Journal | 1 |
| 28 | Journal of Systems Architecture | Journal | 1 |
| 29 | Jurnal Teknologi | Journal | 1 |
| 30 | MDPI-Computers | Journal | 2 |
| 31 | Revista | Journal | 1 |
| 32 | Universal Access in the Information Society | Journal | 1 |
| 33 | User Modeling and User-Adapted Interaction | Journal | 1 |
| 34 | AIS Transactions on Human-Computer Interaction | Journal | 1 |
| 35 | Cognizance Journal of Multidisciplinary Studies | Journal | 1 |
| 36 | MAP Education and Humanities | Journal | 1 |
| 37 | International Journal of Artificial Intelligence in Education | Journal | 1 |
| 38 | British Journal of Educational Technology | Journal | 1 |
| 39 | Jurnal Pendidikan Bahasa Asing Dan Sastra | Journal | 1 |
| 40 | Frontiers in Psychology | Journal | 1 |
| 41 | Computer Assisted Language Learning | Journal | 1 |

| Sr. No. | Publication source | Channel | No. of articles |
|---|---|---|---|
| 42 | International Journal of Gaming and Computer-Mediated Simulations (IJGCMS) | Journal | 1 |
| 43 | Sustainable Social Development | Journal | 1 |
| 44 | International Journal for Multidisciplinary Research (IJFMR) | Journal | 1 |
| 45 | Jurnal Education and Development | Journal | 1 |
| 46 | 2017 40th International Convention on Information and Communication Technology, Electronics and Microelectronics (MIPRO) | Conference | 1 |
| 47 | 2017 6th IIAI International Congress on Advanced Applied Informatics | Conference | 1 |
| 48 | 2017 IEEE Symposium on Visual Languages and Human-CeNtric Computing (VL/HCC) | Conference | 1 |
| 49 | 2018 7th International Congress on Advanced Applied Informatics | Conference | 1 |
| 50 | 2020 IEEE Global Engineering Education Conference | Conference | 1 |
| 51 | 4th International Conference on Computing Sciences | Conference | 1 |
| 52 | 6th Conference on Engineering Education (ICEED) | Conference | 1 |
| 53 | IEEE Global Engineering Education Conference (EDUCON) | Conference | 1 |
| 54 | Interactive Mobile Communication Technologies and Learning: Proceedings of the 11th IMCL Conference | Conference | 1 |
| 55 | Mobile Technologies and Applications for the Internet of Things: Proceedings of the 12th IMCL Conference | Conference | 1 |
| 56 | NordiCHI: Nordic Conference on Human-Computer Interaction | Conference | 1 |
| 57 | Procedia Computer Science | Conference | 1 |
| 58 | Proceedings of the 14th International Conference on the Foundations of Digital Games | Conference | 2 |
| 59 | Proceedings of the 15th International Academic Mindtrek Conference on Envisioning Future Media Environments | Conference | 1 |
| 60 | Proceedings of the 15th International Conference on Computer Systems and Technologies-CompSysTech | Conference | 1 |
| 61 | Proceedings of the 18th ACM International Conference on Interaction Design and Children | Conference | 1 |
| 62 | Proceedings of the 2017 ACM Conference on International Computing Education Research | Conference | 1 |
| 63 | Proceedings of the 2019 ACM Conference on Innovation and Technology in Computer Science Education | Conference | 1 |
| 64 | Proceedings of the 2019 ACM Conference on International Computing Education Research | Conference | 1 |
| 65 | Proceedings of the 50th ACM Technical Symposium on Computer Science Education | Conference | 1 |
| 66 | Proceedings of the 51st Hawaii International Conference on System Sciences | Conference | 1 |
| 67 | Proceedings of the ACM Conference on Global Computing Education | Conference | 1 |
| 68 | Proceedings of the ACM on Human-Computer Interaction, 5(CHI PLAY), | Conference | 1 |
| 69 | Proceedings of the Sixth International Conference on Technological Ecosystems for Enhancing Multiculturality | Conference | 1 |
| 70 | Proceedings of the Working Group Reports on Innovation and Technology in Computer Science Education | Conference | 1 |
| 71 | Systems, Software and Services Process Improvement: 26th European conference, EuroSPI 2019, Edinburgh, UK | Conference | 1 |
| 72 | World Conference on Educational Multimedia, Hypermedia & Telecommunications | Conference | 1 |
| 73 | 8th International Symposium on Telecommunications (IST). | Conference | 1 |
| 74 | GHITALY@CHItaly. 1st Workshop on Games-Human Interaction | Conference | 1 |
| 75 | In Proceedings of the 8th International Conference on Sustainable Information Engineering and Technology, 2023 | Conference | 1 |
| 76 | In 2023 7th IEEE Congress on Information Science and Technology (CiSt) | Conference | 1 |
| 77 | A Design-Based Research Study. arXiv Preprint arXiv:2302.12834, 2023 | Conference | 1 |
| 78 | Proceedings of the III International Conference on Advances in Science, Engineering, and Digital Education: ASEDU-III 2022 | Conference | 1 |

**Table 6** Quality assessment of the selected studies.

| Sr. No. | Ref. | Classification | | | | | Evaluation score | | | | |
|---|---|---|---|---|---|---|---|---|---|---|---|
| | | P. Channel | Publication year | Research type | Empirical type | Methodology | (a) 1 | (b) 2 | (c) 3 | (d) 4 | Score |
| 1 | *Giannakoulas & Xinogalos (2018)* | Journal | 2018 | Evaluation research | Survey | TAM model used | 1 | 1 | 2 | 4 | 8 |
| 2 | *Zatarain Cabada et al. (2020)* | Journal | 2018 | Evaluation research | Experiment | TAM model used | 1 | 1 | 1 | 4 | 7 |
| 3 | *Malliarakis, Satratzemi & Xinogalos (2017)* | Journal | 2016 | Evaluation research | Experiment and survey | Evaluation framework | 1 | 1 | 2 | 4 | 8 |
| 4 | *Papadakis & Kalogiannakis (2019)* | Journal | 2019 | Evaluation research | Experiment and interview | Mix method | 0 | 1 | 2 | 2 | 5 |
| 5 | *Topalli & Cagiltay (2018)* | Journal | 2018 | Solution proposal | Experiment | Questionnaire | 1 | 1 | 2 | 4 | 8 |
| 6 | *Chang, Chung & Chang (2020)* | Journal | 2020 | Solution Proposal | Experiment and survey | Questionnaire | 1 | 1 | 2 | 4 | 8 |
| 7 | *Jakoš & Verber (2017)* | Journal | 2016 | Evaluation research | Experiment and Survey | Questionnaire | 1 | 1 | 2 | 4 | 8 |
| 8 | *Garneli & Chorianopoulos (2018)* | Journal | 2017 | Evaluation research | Experiment | Interview and observation | 1 | 1 | 1 | 4 | 7 |
| 9 | *Mathew, Malik & Tawafak (2019)* | Journal | 2019 | Solution proposal | Survey | Interview | 1 | 1 | 1 | 3 | 6 |
| 10 | *Pellas & Vosinakis (2018)* | Journal | 2018 | Solution proposal | Experiment | Questionnaire | 1 | 1 | 2 | 4 | 8 |
| 11 | *Wei et al. (2021)* | Journal | 2019 | Solution proposal | Experiment and survey | Questionnaire and interview | 1 | 1 | 2 | 4 | 8 |
| 12 | *Strawhacker & Bers (2019)* | Journal | 2018 | Evaluation research | Experiment and survey | Mix method | 1 | 1 | 2 | 4 | 8 |
| 13 | *Hitchens & Tulloch (2018)* | Journal | 2018 | Solution proposal | Survey | Questionnaire | 1 | 1 | 1 | 2 | 5 |
| 14 | *Syaifudin, Funabiki & Kuribayashi (2019)* | Journal | 2019 | Solution proposal | N/A | N/A | 1 | 1 | 1 | 1 | 4 |
| 15 | *Marwan, Jay Williams & Price (2019)* | Conference | 2019 | Evaluation research | Survey | Observation | 1 | 1 | 1 | 2 | 5 |
| 16 | *Maskeliūnas et al. (2020)* | Journal | 2020 | Solution proposal | Experiment and survey | Questionnaire | 1 | 1 | 1 | 2 | 5 |
| 17 | *Duffany (2017)* | Journal | 2017 | Solution proposal | N/A | Observation | 1 | 0 | 1 | 2 | 4 |
| 18 | *Seraj, Autexier & Janssen (2018)* | Conference | 2018 | Solution proposal | N/A | Observation | 1 | 1 | 2 | 2 | 6 |
| 19 | *Figueiredo & García-Peñalvo (2018)* | Conference | 2018 | Evaluation research | Review | Observation | 1 | 0 | 1 | 2 | 4 |
| 20 | *Krugel & Hubwieser (2017)* | Conference | 2017 | Solution proposal | Survey | Questionnaire and interview | 1 | 1 | 2 | 2 | 6 |
| 21 | *Skalka & Drlík (2018)* | Journal | 2017 | Solution proposal | Review | Observation | 1 | 1 | 1 | 2 | 5 |
| 22 | *Nadolny et al. (2017)* | Journal | 2017 | Evaluation research | Survey | Questionnaire | 1 | 1 | 2 | 4 | 8 |

| Sr. No. | Ref. | Classification | | | | | Evaluation score | | | | |
|---|---|---|---|---|---|---|---|---|---|---|---|
| | | P. Channel | Publication year | Research type | Empirical type | Methodology | (a) 1 | (b) 2 | (c) 3 | (d) 4 | Score |
| 23 | *Hooshyar, Yousefi & Lim (2019)* | Journal | 2017 | Solution proposal | Survey | Questionnaire and interview | 1 | 1 | 2 | 4 | 8 |
| 24 | *Von Hausswolff (2017)* | Conference | 2017 | Evaluation research | Survey | N/A | 1 | 0 | 1 | 2 | 4 |
| 25 | *Drosos, Guo & Parnin (2017)* | Conference | 2017 | Solution proposal | Survey | Questionnaire | 1 | 0 | 1 | 2 | 4 |
| 26 | *Bernik, Radošević & Bubaš (2017)* | Conference | 2017 | Solution proposal | Survey | Questionnaire | 1 | 1 | 1 | 2 | 5 |
| 27 | *Troiano et al. (2019)* | Conference | 2019 | Solution proposal | Survey | Questionnaire | 1 | 1 | 1 | 2 | 5 |
| 28 | *Malik et al. (2019)* | Journal | 2019 | Solution proposal | Survey | Questionnaire | 1 | 1 | 2 | 3 | 7 |
| 29 | *Devine et al. (2019)* | Journal | 2019 | Evaluation research | Survey | Questionnaire | 1 | 1 | 2 | 2 | 6 |
| 30 | *Yallihep & Kutlu (2020)* | Journal | 2019 | Evaluation research | Experiment and survey | Questionnaire | 1 | 1 | 2 | 3 | 7 |
| 31 | *Piedade et al. (2020)* | Journal | 2020 | Evaluation research | Mix method | Questionnaire and interview | 1 | 1 | 2 | 3 | 7 |
| 32 | *Luik et al. (2019)* | Journal | 2019 | Evaluation research | Mix method | Questionnaire and interview | 1 | 1 | 2 | 4 | 8 |
| 33 | *Luxton-Reilly et al. (2019)* | Conference | 2019 | Evaluation research | Survey | Questionnaire | 1 | 1 | 1 | 2 | 5 |
| 34 | *Martins, de Almeida Souza Concilio & de Paiva Guimarães (2018)* | Journal | 2018 | Solution proposal | Experiment and survey | Questionnaire | 1 | 1 | 2 | 3 | 7 |
| 35 | *Smith et al. (2019)* | Conference | 2019 | Evaluation research | Survey | Questionnaire | 1 | 1 | 2 | 2 | 6 |
| 36 | *Schez-Sobrino et al. (2020)* | Journal | 2020 | Solution proposal | Survey | Questionnaire | 1 | 1 | 2 | 3 | 7 |
| 37 | *Ivanović et al. (2017)* | Journal | 2016 | Evaluation research | Survey | Questionnaire | 1 | 1 | 2 | 4 | 8 |
| 38 | *Hellings & Haelermans (2020)* | Journal | 2020 | Evaluation research | Experiment and Survey | Questionnaire | 1 | 1 | 2 | 3 | 7 |
| 39 | *Marwan, Jay Williams & Price (2019)* | Conference | 2019 | Evaluation research | Survey | Questionnaire | 1 | 1 | 1 | 2 | 5 |
| 40 | *Laporte & Zaman (2018)* | Conference | 2017 | Evaluation research | N/A | N/A | 1 | 0 | 1 | 2 | 4 |
| 41 | *Kumar & Sharma (2018)* | Conference | 2018 | Solution proposal | N/A | N/A | 1 | 0 | 1 | 2 | 4 |
| 42 | *de Pontes, Guerrero & de Figueiredo (2019)* | Conference | 2019 | Solution proposal | Survey | Questionnaire | 1 | 1 | 2 | 2 | 6 |
| 43 | *Paiva, Leal & Queirós (2020)* | Journal | 2020 | Solution Proposal | Experiment and survey | Questionnaire | 1 | 1 | 2 | 2 | 6 |
| 44 | *Wong & Yatim (2018)* | Conference | 2018 | Solution proposal | Experiment and survey | Questionnaire | 1 | 1 | 1 | 2 | 5 |

*(Continued)*

| Sr. No. | Ref. | Classification | | | | | Evaluation score | | | | |
|---|---|---|---|---|---|---|---|---|---|---|---|
| | | P. Channel | Publication year | Research type | Empirical type | Methodology | (a) 1 | (b) 2 | (c) 3 | (d) 4 | Score |
| 45 | *Gulec et al. (2019)* | Conference | 2019 | Solution proposal | Experiment and survey | Questionnaire | 1 | 1 | 2 | 2 | 6 |
| 46 | *Tasadduq et al. (2021)* | Journal | 2021 | Evaluation research | Survey | Questionnaire | 1 | 1 | 2 | 3 | 7 |
| 47 | *Abbasi et al. (2021)* | Journal | 2021 | Solution proposal | Experiment and survey | Questionnaire | 1 | 1 | 2 | 3 | 7 |
| 48 | *Zhu et al. (2019)* | Conference | 2019 | Solution proposal | Experiment and survey | Questionnaire | 1 | 1 | 2 | 2 | 6 |
| 49 | *Sideris & Xinogalos (2019)* | Journal | 2019 | Solution proposal | Experiment and survey | Questionnaire | 1 | 1 | 2 | 4 | 8 |
| 50 | *Montes et al. (2021)* | Journal | 2021 | Solution proposal | Experiment and Survey | Questionnaire | 1 | 1 | 2 | 4 | 8 |
| 51 | *Xinogalos & Tryfou (2021)* | Journal | 2021 | Solution Proposal | Experiment and survey | Questionnaire | 1 | 1 | 2 | 3 | 7 |
| 52 | *Toukiloglou & Xinogalos (2022)* | Journal | 2022 | Solution proposal | Experiment and survey | Questionnaire | 1 | 1 | 2 | 4 | 8 |
| 53 | *Daungcharone, Panjaburee & Thongkoo (2017)* | Conference | 2017 | Solution proposal | Experiment and survey | Questionnaire | 1 | 1 | 2 | 2 | 6 |
| 54 | *Carreño-León, Rodríguez-Álvarez & Sandoval-Bringas (2019)* | Conference | 2019 | Solution PRoposal | Experiment and survey | Questionnaire | 1 | 1 | 2 | 2 | 6 |
| 55 | *Jemmali et al. (2019)* | Conference | 2019 | Solution proposal | N/A | N/A | 1 | 0 | 1 | 2 | 4 |
| 56 | *Khaleel, Ashaari & Wook (2019)* | Journal | 2019 | Solution proposal | Experiment and Survey | Questionnaire | 1 | 1 | 2 | 2 | 6 |
| 57 | *Moreno & Pineda (2018)* | Journal | 2018 | Evaluation Research | N/A | N/A | 1 | 0 | 1 | 2 | 4 |
| 58 | *Sanmugam, Abdullah & Zaid (2014)* | Conference | 2016 | Evaluation research | N/A | N/A | 1 | 1 | 1 | 2 | 5 |
| 59 | *Pankiewicz (2020)* | Conference | 2020 | Evaluation research | Survey | Questionnaire | 1 | 1 | 1 | 2 | 5 |
| 60 | *Queirós (2019)* | Journal | 2019 | Solution proposal | Experiment and survey | Questionnaire | 1 | 1 | 1 | 2 | 5 |
| 61 | *Azmi, Iahad & Ahmad (2016)* | Journal | 2016 | Evaluation research | Survey | Questionnaire | 1 | 1 | 1 | 2 | 5 |
| 62 | *Bennani, Maalel & Ghezala (2020)* | Journal | 2020 | Evaluation research | Survey | N/A | 1 | 1 | 1 | 2 | 5 |
| 63 | *Challco et al. (2016)* | Journal | 2016 | Solution Proposal | Experiment and Survey | Questionnaire | 1 | 1 | 2 | 4 | 8 |
| 64 | *De-Marcos, Garcia-Lopez & Garcia-Cabot (2016)* | Journal | 2016 | Solution proposal | Experiment and survey | Questionnaire | 1 | 1 | 2 | 4 | 8 |
| 65 | *Deterding (2016)* | Conference | 2016 | Evaluation research | Survey | Questionnaire | 1 | 1 | 1 | 2 | 5 |
| 66 | *Hassan et al. (2021)* | Journal | 2019 | Evaluation research | Mix method | Questionnaire and interviews | 1 | 1 | 2 | 4 | 8 |

| Sr. No. | Ref. | Classification | | | | | Evaluation score | | | | |
|---|---|---|---|---|---|---|---|---|---|---|---|
| | | P. Channel | Publication year | Research type | Empirical type | Methodology | (a) 1 | (b) 2 | (c) 3 | (d) 4 | Score |
| 67 | *Katan & Anstead (2020)* | Conference | 2020 | Evaluation research | Survey | Questionnaire | 1 | 1 | 1 | 2 | 5 |
| 68 | *Kiraly & Balla (2020)* | Journal | 2020 | Evaluation research | Survey | Questionnaire | 1 | 1 | 1 | 2 | 5 |
| 69 | *Knutas et al. (2016)* | Conference | 2016 | Evaluation research | Survey | Questionnaire | 1 | 1 | 1 | 2 | 5 |
| 70 | *Knutas et al. (2017)* | Conference | 2017 | Evaluation research | Survey | Questionnaire | 1 | 1 | 1 | 2 | 5 |
| 71 | *Knutas et al. (2019)* | Journal | 2018 | Solution proposal | Experiment and survey | Questionnaire | 1 | 1 | 2 | 4 | 8 |
| 72 | *Marín et al. (2019)* | Journal | 2019 | Solution proposal | Experiment and survey | Questionnaire | 1 | 1 | 2 | 4 | 8 |
| 73 | *Mullins & Sabherwal (2018)* | Conference | 2018 | Evaluation research | Survey | Questionnaire | 1 | 0 | 1 | 2 | 4 |
| 74 | *Mullins & Sabherwal (2020)* | Journal | 2020 | Evaluation research | Survey | N/A | 1 | 0 | 1 | 2 | 4 |
| 75 | *de Marcos Ortega, García-Cabo & López (2017)* | Journal | 2017 | Evaluation research | Survey | Questionnaire | 1 | 1 | 1 | 2 | 5 |
| 76 | *Rodrigues et al. (2021)* | Conference | 2021 | Evaluation research | Survey | Questionnaire | 1 | 1 | 1 | 2 | 5 |
| 77 | *Rodrigues et al. (2022)* | Journal | 2022 | Solution proposal | Experiment and survey | Questionnaire | 1 | 1 | 2 | 4 | 8 |
| 78 | *Rojas-López et al. (2019)* | Journal | 2019 | Solution proposal | Experiment and survey | Questionnaire | 1 | 1 | 1 | 3 | 6 |
| 79 | *Roosta, Taghiyareh & Mosharraf (2016)* | Conference | 2016 | Evaluation research | Survey | N/A | 1 | 0 | 1 | 2 | 4 |
| 80 | *Santos et al. (2021)* | Journal | 2021 | Solution proposal | Experiment and survey | Questionnaire | 1 | 1 | 2 | 3 | 7 |
| 81 | *Toda et al. (2019)* | Journal | 2019 | Evaluation research | Survey | Questionnaire | 1 | 1 | 2 | 3 | 7 |
| 82 | *Hong, Saab & Admiraal (2024)* | Journal | 2024 | Evaluation research | Review | N/A | 1 | 0 | 2 | 4 | 7 |
| 83 | *Pradana et al. (2023)* | Conference | 2023 | Evaluation research | Experiment and survey | Quasi-experiment | 1 | 1 | 1 | 1 | 4 |
| 84 | *Zhang & Hasim (2023)* | Journal | 2023 | Evaluation research | Review | N/A | 1 | 1 | 1 | 3 | 6 |
| 85 | *Permana, Permatawati & Khoerudin (2023)* | Journal | 2023 | | Review | Questionnaire and interviews | 1 | 1 | 2 | 2 | 6 |
| 86 | *Dehghanzadeh et al. (2024)* | Journal | 2024 | Evaluation research | Review | N/A | 1 | 1 | 1 | 4 | 8 |
| 87 | *Shortt et al. (2023)* | Journal | 2022 | Evaluation research | Review | N/A | 1 | 1 | 2 | 4 | 8 |
| 88 | *Huseinović (2024)* | Journal | 2024 | Evaluation research | Experiment and survey | Questionnaire | 1 | 1 | 1 | 0 | 3 |

(Continued)

| Sr. No. | Ref. | Classification | | | | | Evaluation score | | | | |
|---|---|---|---|---|---|---|---|---|---|---|---|
| | | P. Channel | Publication year | Research type | Empirical type | Methodology | (a) 1 | (b) 2 | (c) 3 | (d) 4 | Score |
| 89 | *Rodrigues et al. (2023)* | Journal | 2023 | Evaluation research | Experiment and survey | N/A | 1 | 1 | 2 | 4 | 8 |
| 90 | *Amer et al. (2023)* | Journal | 2023 | Evaluation research | Experiment and survey | Interviews | 1 | 1 | 2 | 1 | 5 |
| 91 | *Gharbaoui, Mansouri & Poirier (2023)* | Conference | 2023 | Solution proposal | Experiment and survey | Questionnaire | 1 | 0 | 1 | 2 | 4 |
| 92 | *Scamardella, Saraiello & Tafuri (2023)* | Journal | 2023 | Evaluation research | N/A | N/A | 1 | 1 | 1 | 2 | 5 |
| 93 | *Kitani (2023)* | Journal | 2023 | Solution proposal | Experiment and survey | Interviews | 1 | 1 | 1 | 2 | 5 |
| 94 | *Ghosh & Pramanik (2023)* | Journal | 2023 | Evaluation research | Experiment and survey | Questionnaire | 1 | 1 | 1 | 2 | 5 |
| 95 | *Cao (2023)* | N/A | 2023 | Evaluation research | Experiment and survey | Questionnaire | 1 | 1 | 2 | 1 | 5 |
| 96 | *Zourmpakis, Kalogiannakis & Papadakis (2023)* | Journal | 2023 | Solution proposal | Experiment and survey | Questionnaire | 1 | 1 | 2 | 3 | 7 |
| 97 | *Oliveira et al. (2023)* | Journal | 2023 | Evaluation research | Review | N/A | 1 | 1 | 2 | 4 | 8 |
| 98 | *Rakhmanita, Kusumawardhani & Anggarini (2023)* | Journal | 2023 | Evaluation research | Mix method | Observation and interview | 1 | 1 | 1 | 3 | 6 |
| 99 | *Janson et al. (2023)* | Journal | 2023 | Evaluation research | N/A | N/A | 1 | 0 | 1 | 2 | 4 |
| 100 | *Suresh Babu & Dhakshina Moorthy (2024)* | Journal | 2024 | Evaluation research | Review | Observation | 1 | 1 | 2 | 4 | 8 |
| 101 | *Dehghanzadeh et al. (2024)* | Journal | 2024 | Evaluation research | Review | N/A | 1 | 1 | 2 | 3 | 7 |

Further, types of research have been categorized as Evaluation framework, Evaluation research, Solution proposal, and Review. All studies have empirically validated their results by performing statistical analysis, experiments, surveys, or case studies to increase their quality standards, awarded one score each. In category (c) of quality assessment criteria, only 11 out of 81 studies have not presented an empirical result that was awarded a zero score. In contrast, no study scored zero for category (d) of quality assessment criteria, but forty-five (45) studies got the lowest score in the same section. In addition, Table 7 presents the total studies that secure the highest to lowest scores accordingly.

## RQ3: What are the prevailing trends and optimal methodologies for integrating personalized gamification frameworks in programming education, and what distinctions exist in the design and customization of these frameworks?

Personalized gamified programming education has emerged as an innovative and engaging approach to enhancing students' learning experiences. However, designing an effective

**Table 7 Accumulative quality assessment score.**

| References | Score | Total |
|---|---|---|
| Challco et al. (2016), Chang, Chung & Chang (2020), De-Marcos, Garcia-Lopez & Garcia-Cabot (2016), Giannakoulas & Xinogalos (2018), Hassan et al. (2021), Hooshyar, Yousefi & Lim (2019), Ivanović et al. (2017), Jakoš & Verber (2017), Knutas et al. (2019), Luik et al. (2019), Malliarakis, Satratzemi & Xinogalos (2017), Marín et al. (2019), Montes et al. (2021), Nadolny et al. (2017), Pellas & Vosinakis (2018), Rodrigues et al. (2022), Sideris & Xinogalos (2019), Strawhacker & Bers (2019), Topalli & Cagiltay (2018), Toukiloglou & Xinogalos (2022), Wei et al. (2021), Dehghanzadeh et al. (2024), Shortt et al. (2023), Rodrigues et al. (2023), Oliveira et al. (2023), Suresh Babu & Dhakshina Moorthy (2024) | 8 | 26 |
| Abbasi et al. (2021), Dehghanzadeh et al. (2024), Garneli & Chorianopoulos (2018), Hellings & Haelermans (2020), Malik et al. (2019), Martins, de Almeida Souza Concilio & de Paiva Guimarães (2018), Piedade et al. (2020), Santos et al. (2021), Schez-Sobrino et al. (2020), Toda et al. (2019), Tasadduq et al. (2021), Xinogalos & Tryfou (2021), Yallihep & Kutlu (2020), Zatarain Cabada et al. (2020), Hong, Saab & Admiraal (2024), Zourmpakis, Kalogiannakis & Papadakis (2023) | 7 | 16 |
| Carreño-León, Rodríguez-Álvarez & Sandoval-Bringas (2019), Daungcharone, Panjaburee & Thongkoo (2017), de Pontes, Guerrero & de Figueiredo (2019), Devine et al. (2019), Gulec et al. (2019), Khaleel, Ashaari & Wook (2019), Krugel & Hubwieser (2017), Mathew, Malik & Tawafak (2019), Paiva, Leal & Queirós (2020), Rojas-López et al. (2019), Seraj, Autexier & Janssen (2018), Smith et al. (2019), Zhu et al. (2019), Zhang & Hasim (2023), Permana, Permatawati & Khoerudin (2023), Rakhmanita, Kusumawardhani & Anggarini (2023) | 6 | 16 |
| Azmi, Iahad & Ahmad (2016), Bennani, Maalel & Ghezala (2020), Bernik, Radošević & Bubaš (2017), de Marcos Ortega, García-Cabo & López (2017), Deterding (2016), Hitchens & Tulloch (2018), Katan & Anstead (2020), Kiraly & Balla (2020), Knutas et al. (2016, 2017), Luxton-Reilly et al. (2019), Marwan, Jay Williams & Price (2019), Maskeliūnas et al. (2020), Pankiewicz (2020), Papadakis & Kalogiannakis (2019), Queirós (2019), Rodrigues et al. (2021), Sanmugam, Abdullah & Zaid (2014), Skalka & Drlík (2018), Troiano et al. (2019), Wong & Yatim (2018), Amer et al. (2023) | 5 | 23 |
| Arkhipova et al. (2024), Drosos, Guo & Parnin (2017), Duffany (2017), Figueiredo & García-Peñalvo (2018), Jemmali et al. (2019), Kumar & Sharma (2018), Laporte & Zaman (2018), Moreno & Pineda (2018), Mullins & Sabherwal (2018, 2020), Roosta, Taghiyareh & Mosharraf (2016), Syaifudin, Funabiki & Kuribayashi (2019), Von Hausswolff (2017), Pradana et al. (2023), Gharbaoui, Mansouri & Poirier (2023), Janson et al. (2023) | 4 | 16 |
| Huseinović (2024), Scamardella, Saraiello & Tafuri (2023), Kitani (2023) | 3 | 3 |

**Table 8 Summary of gamification frameworks used in programming language education.**

| Gamification framework | Number of studies |
|---|---|
| Adopted frameworks: | |
| Technology Acceptance Model (TAM) | 1 |
| Attention, Relevance, Confidence, and Satisfaction (ARCS) | 2 |
| TETEM | 1 |
| Custom frameworks | 17 |
| Not Specified (NS) | 15 |
| GBL | 1 |
| Sokoon | 1 |
| Gamified mobile-assisted language learning application | 1 |

personalized gamified programming education system requires a deep understanding of the relevant theories and frameworks/conceptual models used in this context. This section aims to identify the frameworks/conceptual models that have been applied to personalized gamified programming education with respect to students' cognition research and explore the relationships between them. This section will provide a comprehensive overview of the theoretical foundations underpinning personalized gamified programming education,

which can serve as a valuable resource for researchers and educators in this field. This research question will explore using adopted and custom frameworks in gamification for programming education. In our analysis, several articles did not mention any specific gamification framework. We referred to them as 'Not specified' (NS). Portions of this text were previously published as part of a preprint (*Ishaq & Alvi, 2023*).

Table 8 summarizes the studies that used each gamification framework for programming language education interventions. The "Adopted Frameworks" category includes previously developed and used in other contexts, while the "Custom Frameworks" category includes frameworks specifically designed for the intervention. The "Not Specified (NS)" category includes studies that did not explicitly mention using any gamification framework.

**Adopted gamification frameworks:** Previous research has used various gamification frameworks such as the Technology Acceptance Model (TAM), the ARCS model, and the Turkish Educational Technology Evaluation Model (TETEM). The TAM framework investigates learners' acceptance of gamification elements, while the ARCS model aims to motivate learners by drawing their attention to the material, emphasizing its relevance, building their confidence, and providing satisfaction and rewards. TETEM is a framework used to evaluate educational technologies in the Turkish context and has been used in several research studies. Despite being popular, only two articles used ARCS in the context of gamification for programming language education. Among the 36 articles reviewed, only two studies used frameworks such as TAM and TETEM, while ARCS was used in two articles. For instance, *Zatarain Cabada et al. (2020)* employed the TAM framework to evaluate students' perceptions of automated programming hints. Another study by *Maskeliūnas et al. (2020)* used both TAM and TETEM to assess the effectiveness of an interactive mobile game for learning programming. *Khaleel, Ashaari & Wook (2019)* used the ARCS model to guide the design of a gamified learning system to improve learning outcomes in a programming language website.

**Custom gamification frameworks:** Gamification techniques have been explored in several studies to enhance programming education. Custom gamification frameworks provide greater control over the design and implementation of gamification techniques, but their development can be resource-intensive and require high technical expertise. Some studies have used custom frameworks, such as CMX (*Malliarakis, Satratzemi & Xinogalos, 2017*), a microlearning-based mobile application (*Skalka & Drlík, 2018*), and a game-based Bayesian intelligent tutoring system (*Hooshyar et al., 2018*). *Kumar & Sharma (2019)* demonstrated improved student engagement and learning outcomes through a gamified approach that used Bayesian networks as a decision-making tool. *de Pontes, Guerrero & de Figueiredo (2019)* and *Paiva, Leal & Queirós (2020)* implemented frameworks that provided platforms for programming exercises and assessments, both of which improved student engagement and learning outcomes. *Tasadduq et al. (2021)* utilized a custom framework to evaluate the impact of gamification on students with a background in rote learning who are learning computer programming. *Abbasi et al. (2021)*

investigated the effectiveness of serious games in enhancing students' learning performance and motivation using a custom framework. Several articles investigate the use of serious games and gamification in programming education. *Zhu et al. (2019)* utilize a serious game framework to teach parallel programming, while *Sideris & Xinogalos (2019)* present a framework that teaches programming concepts through a 2D platform game. *Xinogalos & Tryfou (2021)* discuss using Greenfoot as a tool for creating serious games for programming education, and *Daungcharone, Panjaburee & Thongkoo (2017)* describe a gaming framework that employs a digital game as a compiler to motivate C programming language learning in higher education. *Carreño-León, Rodríguez-Álvarez & Sandoval-Bringas (2019)* introduce a gaming framework that uses gamification techniques to enhance problem-solving skills in programming education, incorporating tailored challenges, a scoring system, and a feedback mechanism to increase student engagement and motivation. Lastly, *Marín et al. (2019)* conducted an empirical study on the effectiveness of gamification techniques in programming courses, incorporating social gamification elements.

**Without gamification frameworks:** Approximately 15 articles reviewed did not explicitly mention using gamification frameworks for programming education. *Topalli & Cagiltay (2018)* proposed using a problem-based learning approach to encourage collaborative game development without a specific framework. *Mathew, Malik & Tawafak (2019)* utilized an educational game called PROSOLVE, incorporating problem-based learning and gamification techniques. *Hitchens & Tulloch (2018)* designed a gamification approach for classroom instruction, integrating game elements such as rewards, feedback, and progress tracking, but did not mention a specific framework. The authors use game design principles such as immediate feedback and gradual increase in difficulty levels to design activities that include badges, points, and leaderboards. Various studies have explored different approaches to programming language education. While some have used specific gamification frameworks, others have not. For instance, *Syaifudin et al. (2020)* proposed an Android Programming Learning Assistant System (APLAS) to help students learn basic Android application development, while *Marwan, Jay Williams & Price (2019)* used a problem-based learning approach and automated programming hints to improve students' performance in programming. Additionally, some studies have focused on active learning techniques such as pair programming and think-pair-share, like the work of *Duffany (2017)*, while others have explored the use of educational robotics, such as *Piedade et al. (2020)*. Furthermore, some studies, like *Luik et al. (2019)*, did not discuss any explicit gaming aspect. However, incorporating gaming aspects such as challenges, points, levels, and feedback can enhance the learning experience in programming language education, as suggested by various studies.

**Game-based learning:** Gamification frameworks are commonly used in programming language education interventions to enhance learner engagement and motivation. However, game-based learning (GBL), which involves using games for learning, is another approach that has been used rather than applying gamification elements to a non-game

context. In programming language education, GBL typically involves designing games or game-like activities that require learners to apply programming concepts to progress or succeed. One example of GBL in programming language education is a mobile application developed by *Chang, Chung & Chang (2020)*, which incorporated game elements such as points, badges, and leaderboards to enhance learner engagement and motivation.

**Validation of framework:** Based on our analysis of available information, we found that some gamification frameworks have been validated, often through structural equation modeling (SEM) or questionnaires. However, the success of a framework depends on various factors, and careful consideration and ongoing evaluation are necessary when adopting or customizing a framework for a specific purpose. It is also important to note that many studies reviewed did not explicitly mention a framework, making it difficult to compare the effectiveness of different interventions.

Table 9 summarizes the references and types of gamification frameworks used in programming language education interventions.

Our literature review identified effective practices and trends in personalized gamification frameworks for programming education. Personalization increased student motivation and engagement, and four categories of articles were identified: those adopting established frameworks, customized frameworks, game elements, and those combining gamification and game-based learning. Educators and instructional designers can use these insights to create effective and engaging learning experiences. However, challenges such as clear goal setting and potential distraction from learning objectives were also identified. Further research is needed to examine the effectiveness of different personalized gamification frameworks in different contexts, and empirical studies are needed to evaluate their effectiveness in real-world settings.

### RQ4: How might personalized gamification frameworks in programming education be correlated with the various cognitive domains delineated within Bloom's taxonomy?

This research question explores the effective alignment of frameworks with different cognitive levels of Bloom's taxonomy to understand how personalized gamification can enhance learning outcomes in programming education. This involves identifying the levels of learning in Bloom's taxonomy and analyzing how they relate to gamification framework design. Bloom's Taxonomy categorizes educational goals into different levels of cognitive complexity. These levels range from lower-order thinking skills, such as remembering and understanding, to higher-order thinking skills, such as synthesizing and evaluating complex information and ideas. The categories of Bloom's Taxonomy can be divided into low-level and high-level thinking skills. Portions of this text were previously published as part of a preprint (*Ishaq & Alvi, 2023*).

**Low-level thinking skills (LL):**

- *Remembering:* recalling facts, information, or procedures.
- *Understanding:* comprehending the meaning of information, including identifying patterns and relationships.

**Table 9 Gamification frameworks used in programming language education.**

| References | Adopted/ Custom | Based on |
|---|---|---|
| *Zatarain Cabada et al. (2020)* | Adopted | TAM |
| *Maskeliūnas et al. (2020)* | Adopted | TAM and TETEM |
| *Wong & Yatim (2018)* | Adopted | ARCS |
| *Khaleel, Ashaari & Wook (2019)* | Adopted | ARCS |
| *Malliarakis, Satratzemi & Xinogalos (2017)* | Custom | Constructivist learning theory |
| *Skalka & Drlík (2018), Kumar & Sharma (2019), de Pontes, Guerrero & de Figueiredo (2019), Tasadduq et al. (2021), Zhu et al. (2019), Daungcharone, Panjaburee & Thongkoo (2017), Carreño-León, Rodríguez-Álvarez & Sandoval-Bringas (2019), Marín et al. (2019)* | Custom | No mention |
| *Hooshyar et al. (2018)* | Custom | Bayesian network based |
| *Paiva, Leal & Queirós (2020)* | Custom | Asura |
| *Abbasi et al. (2021)* | Custom | SG model |
| *Sideris & Xinogalos (2019)* | Custom | Educational Games Design Model proposed by *Ibrahim & Jaafar (2009)* |
| *Xinogalos & Tryfou (2021)* | Custom | Serious Game Design Assessment (SGDA) Framework was created by Mitgutsch and Alvarado. |
| *Topalli & Cagiltay (2018), Hitchens & Tulloch (2018), Syaifudin, Funabiki & Kuribayashi (2019), Duffany (2017), Figueiredo & García-Peñalvo (2018), Skalka & Drlík (2018), Malik et al. (2019), Piedade et al. (2020), Luik et al. (2019), Simon, Geldreich & Hubwieser (2019), Hellings & Haelermans (2022), De-Marcos, Garcia-Lopez & Garcia-Cabot (2016), Benick, Dörrenbächer-Ulrich & Perels (2018)* | NS | |
| *Mathew, Malik & Tawafak (2019)* | NS | They incorporate elements of problem-based learning. |
| *Marwan, Jay Williams & Price (2019)* | NS | Does use a problem-based learning |
| *Chang, Chung & Chang (2020)* | | They used the GBL design model proposed by *Shi & Shih (2015)*. |
| *Ghosh & Pramanik (2023)* | Mix | They discussed game-based challenges and quests, CodeCombat, Codecademy, and Blockly Games and provided interactive and gamified coding environment. |
| *Cao (2023)* | Custom | They designed a prototype of a story-based gamification Intelligent Tutoring System (ITS) in the CS1 course for Chinese students. |

The lower levels of the taxonomy (knowledge, comprehension, and application) involve basic cognitive processes such as memorization, understanding, and application of information.

**High-level thinking skills (HL):**

- *Applying:* using knowledge and skills to solve problems or complete tasks in new situations.
- *Analyzing:* breaking down complex information into smaller parts to better understand it.
- *Evaluating:* making judgments about the value or quality of information or ideas.
- *Creating:* combining knowledge and skills to create something new or original.

**Table 10 Bloom taxonomy alignment.**

| Reference | Bloom Taxonomy–level |
|---|---|
| *Malliarakis, Satratzemi & Xinogalos (2017)*, *Topalli & Cagiltay (2018)*, *Chang, Chung & Chang (2020)*, *Mathew, Malik & Tawafak (2019)*, *Hitchens & Tulloch (2018)*, *Syaifudin, Funabiki & Kuribayashi (2019)*, *Marwan, Jay Williams & Price (2019)*, *Duffany (2017)*, *Figueiredo & García-Peñalvo (2018)*, *Skalka & Drlík (2018)*, *Hooshyar et al. (2018)*, *Malik et al. (2019)*, *Piedade et al. (2020)*, *Luik et al. (2019)*, *Wong & Yatim (2018)*, *Tasadduq et al. (2021)*, *Abbasi et al. (2021)*, *Zhu et al. (2019)*, *Sideris & Xinogalos (2019)*, *Xinogalos & Tryfou (2021)*, *Daungcharone, Panjaburee & Thongkoo (2017)*, *Carreño-León, Rodríguez-Álvarez & Sandoval-Bringas (2019)*, *Khaleel, Ashaari & Wook (2019)*, *De-Marcos, Garcia-Lopez & Garcia-Cabot (2016)*, *Marín et al. (2019)*, *Pradana et al. (2023)*, *Huseinović (2024)*, *Amer et al. (2023)* | HL |
| *Maskeliūnas et al. (2020)*, *Rodrigues et al. (2023)*, *Cao (2023)*, *Zourmpakis, Kalogiannakis & Papadakis (2023)* | LL |
| *Zatarain Cabada et al. (2020)*, *Hellings & Haelermans (2020)*, *Kumar & Sharma (2019)*, *de Pontes, Guerrero & de Figueiredo (2019)*, *Paiva, Leal & Queirós (2020)*, *Benick, Dörrenbächer-Ulrich & Perels (2018)*, *Permana, Permatawati & Khoerudin (2023)*, *Kitani (2023)*, *Ghosh & Pramanik (2023)*, *Gharbaoui, Mansouri & Poirier (2023)* | LL and HL |
| *Simon, Geldreich & Hubwieser (2019)*, *Scamardella, Saraiello & Tafuri (2023)*, *Janson et al. (2023)*, *Rakhmanita, Kusumawardhani & Anggarini (2023)* | No alignment with Bloom |

The higher levels of the taxonomy (analysis, synthesis, and evaluation) involve more complex cognitive processes such as breaking down information into parts, combining ideas to form a new whole, and making judgments about the value or quality of information. Our article's analysis classified them into high-level (HL) and low-level (LL) Bloom taxonomy categories. Out of the 34 articles analyzed, 26 aligned with high-level thinking, one with low-level thinking, six aligned with both high and low-level thinking, and one did not align with either category. For articles where the taxonomy was not clearly stated, we inferred the level of thinking based on the study's outcome. Table 10 presents the references of the 34 articles analyzed in this study and their alignment with Bloom's taxonomy. The articles were categorized as high level (HL), low level (LL), both high and low level (HL and LL), or no alignment based on their focus on either higher-order thinking skills or foundational knowledge. Our literature review found that gamification can increase student engagement and motivation in programming education, especially when using Bloom's taxonomy to design activities that enhance cognitive complexity. The effectiveness of gamification in achieving learning outcomes depends on factors such as the specific outcomes being targeted and the design of the activity. Our findings suggest that gamification and Bloom's taxonomy can positively impact motivation, cognitive complexity, and learning outcomes, providing important insights for educators and instructional designers.

## Gamification aspect and bloom taxonomy

This subsection explores the relationship between gamification activities in programming education and Bloom's Taxonomy. Specifically, we examine how gamification activities align with different levels of cognitive complexity and promote higher-order thinking skills. By analyzing the gamification aspect with Bloom's Taxonomy, this subsection provides important insights into designing effective and engaging learning activities that promote higher-order thinking skills and support programming education. Specifically, we focus on several key gamification aspects, including:

1) *Intrinsic motivation* is the internal drive to engage in a task or activity because it is personally rewarding or satisfying. This type of motivation can be aligned with the higher levels of Bloom's taxonomy, specifically the levels of evaluating and creating. It can help learners stay engaged and motivated during these challenging tasks, as they derive enjoyment and satisfaction from the learning process.

2) *Extrinsic motivation* is driven by external rewards or consequences and aligns with the lower levels of Bloom's taxonomy. It helps learners stay motivated by providing a goal to achieve or avoid negative consequences.

3) *Performance gain* aligns with Bloom's taxonomy levels of applying and analyzing knowledge and skills, where learners are expected to use their knowledge to solve problems and complete tasks effectively. It is measured by how well learners can apply acquired knowledge and skills in real-world situations, such as writing functional code in a programming course.

4) *Attention and engagement* are important prerequisites for effective memory encoding and retrieval, and they can be aligned with Bloom's first level of remembering. Engaged and attentive learners are more likely to process information deeply and form strong memory representations, which can be retrieved later when needed.

5) *Feedback and assessment:* Gamification provides learners constructive feedback on their progress and performance. It aligns with Bloom's taxonomy's higher levels. Feedback helps learners identify errors or gaps in their understanding, while assessment evaluates learners' ability to judge the value, quality, or effectiveness of ideas, products, or solutions.

6) *Collaboration and social learning:* Gamification facilitates collaboration and social learning among learners, supporting the development of higher-level thinking skills.

7) *Creativity and innovation:* It aligns with Bloom's taxonomy by promoting creativity and innovation at the highest level of cognitive taxonomy. It encourages learners to use their imagination and problem-solving skills through engaging and challenging activities.

In analyzing 34 articles, we found that intrinsic motivation in gamification aligns with Bloom's creating and evaluating categories, while extrinsic motivation aligns with applying and analyzing. However, gamification literature has no clear distinction between the two types of motivation. The relationship between motivation and Bloom's taxonomy may vary based on gamification's specific context and application. Though studies have taken different approaches, aligning gamification with Bloom's categories can offer useful insights for incorporating it into educational settings.

### RQ5. What tools and software applications are developed based on personalized gamification frameworks in programming education, and how are these tools tailored to specific programming languages and concepts?

This section identified several tools and software applications developed based on personalized gamification frameworks in programming education. The results are

**Table 11 Tool and applications.**

| Reference | Topic/Concept | Gamification framework | Tool/Software application |
|---|---|---|---|
| *Zatarain Cabada et al. (2020)* | Algorithm and code construction | Adopted | EasyLogic |
| *Malliarakis, Satratzemi & Xinogalos (2017)* | General programming concepts | Custom | CMX environment |
| *Topalli & Cagiltay (2018)* | Introduction to programming course | NS | Gamification of exercises-physical |
| *Chang, Chung & Chang (2020)* | Introductory course | GBL | Programmer Adventure land |
| *Mathew, Malik & Tawafak (2019)* | Introductory programming course | NS | PROSOLVE game based on pseudo-code technique. |
| *Hitchens & Tulloch (2018)* | No topic mentioned in the article | NS | Classroom activities and associated software were designed and implemented |
| *Syaifudin, Funabiki & Kuribayashi (2019)* | Java | NS | Android programming learning assistance system, namely APLAS. |
| *Marwan, Jay Williams & Price (2019)* | General programming concepts | NS | They used iSnap and added hints to it. |
| *Maskeliūnas et al. (2020)* | Javascript | NS | They developed a game. No name is mentioned. |
| *Duffany (2017)* | Visual Basic | NS | The classroom activities were designed to support active learning |
| *Figueiredo & García-Peñalvo (2018)* | To program mobile robots, microcontrollers, and smart environments | NS | Block-based Enduser programming tool |
| *Skalka & Drlík (2018)* | Introduction to computational thinking and object-oriented concepts | NS | MOOC called LOOP (Learning Object-oriented Programming) |
| *Skalka & Drlík (2018)* | Programming concept | Custom | Only framework is proposed |
| *Hooshyar et al. (2018)* | Introduction to programming course | Custom | Online game-based bayesian intelligent tutoring system (OGITS) |
| *Benick, Dörrenbächer-Ulrich & Perels (2018)* | Batch and Stack | NS | They used Moodle with Gamification features |
| *Malik et al. (2019)* | Introductory programming (IP) courses | NS | PROBSOL |
| *Piedade et al. (2020)* | Programming fundamentals | NS | Not mentioned |
| *Kumar & Sharma (2019)* | Programming concepts | Custom | Development of ProLounge (Programming Lounge)—an online learning application. |
| *de Pontes, Guerrero & de Figueiredo (2019)* | Introductory Programming course | Custom | They designed a game |
| *Paiva, Leal & Queirós (2020)* | Game-based programming challenges (Java) | Custom | ASURA |
| *Wong & Yatim (2018)* | OOP | Adopted | Odyssey of Phoenix |
| *Tasadduq et al. (2021)* | C | Custom | CYourWay |
| *Abbasi et al. (2021)* | OOP | Custom | 2D game named as Object Oriented serious game (OOsg) |
| *Zhu et al. (2019)* | Concurrent and parallel programming (CPP) skills | Custom | Parallel |
| *Sideris & Xinogalos (2019)* | Programming concepts using Python | Custom | PY-RATE ADVENTURES |
| *Xinogalos & Tryfou (2021)* | OOP | Custom | Game of Code: Lost in Javaland |
| *Daungcharone, Panjaburee & Thongkoo (2017)* | C | Custom | A digital game named CPGame |

| Reference | Topic/Concept | Gamification framework | Tool/Software application |
|---|---|---|---|
| *Carreño-León, Rodríguez-Álvarez & Sandoval-Bringas (2019)* | Introductory programming course | Custom | No tool was developed |
| *Khaleel, Ashaari & Wook (2019)* | OOP (Java) | Adopted | Gamified website was developed |
| *Marín et al. (2019)* | C | Custom | A gamified platform, namely UDPiler |
| *Cao (2023)* | Introductory programming course | Custom | Story-based gamified Intelligent Tutoring Systems (ITS) was developed |
| *Pradana et al. (2023)* | HTML, CSS | Adopted | HSS gamification platform |

summarized in Table 11. These tools incorporate game elements and mechanics into programming tasks to improve student motivation, engagement, and learning outcomes. The frameworks used to develop these tools vary. Despite differences in frameworks, the tools share common features such as badges, points, leaderboards, and rewards to incentivize student performance. Most tools are tailored to specific programming languages and concepts and provide personalized feedback and adaptive challenges to meet individual learner needs. Our review suggests personalized gamification can effectively enhance student motivation, engagement, and learning outcomes in programming education. The development of tailored tools and software applications that align with specific programming languages and concepts can further enhance the effectiveness of gamification in programming education. However, more research is needed to evaluate the long-term effects of these tools on student learning outcomes and to identify best practices for designing and implementing gamified programming education tools. Portions of this text were previously published as part of a preprint (*Ishaq & Alvi, 2023*).

## RQ6: What are the common processes, tools, and instruments utilized for evaluating applications based on personalized gamified programming education? What evaluation measures are employed to assess applications from various viewpoints, such as teaching, learning, and technical perspectives?

The methodology refers to the fundamental techniques or methods used to identify, collect, retrieve, and interpret information on the topic (*Paul, 2000*). This research question posed to examine the tools and evaluation methodologies by the selected studies is presented in Tables 12 and 13, whereas Table 12 shows that 37 out of 81 studies used quantitative research methodology by asking the questions from participants in a questionnaire/survey. This research question explored the tools and evaluation methodologies used by the selected studies. Table 13 presents the findings that 37 out of 81 studies utilized quantitative research methodology by administering questionnaires or surveys to participants. According to the evaluation tools, 33 studies, which is the majority, used Statistical Package for Social Science (SPSS) to evaluate the data accordingly, whereas

**Table 12 Methodology adopted by the studies.**

| Source Ref. | Methodology | Total |
|---|---|---|
| *Abbasi et al. (2021), Carreño-León, Rodríguez-Álvarez & Sandoval-Bringas (2019), Chang, Chung & Chang (2020), Daungcharone, Panjaburee & Thongkoo (2017), Giannakoulas & Xinogalos (2018), Garneli & Chorianopoulos (2018), Gulec et al. (2019), Hellings & Haelermans (2020), Ivanović et al. (2017), Jakoš & Verber (2017), Khaleel, Ashaari & Wook (2019), Kumar & Sharma (2018), Malliarakis, Satratzemi & Xinogalos (2017), Marín et al. (2019), Martins, de Almeida Souza Concilio & de Paiva Guimarães (2018), Marwan, Jay Williams & Price (2019), Mathew, Malik & Tawafak (2019), Montes et al. (2021), Moreno & Pineda (2018), Paiva, Leal & Queirós (2020), Pankiewicz (2020), Papadakis & Kalogiannakis (2019), Pellas & Vosinakis (2018), Schez-Sobrino et al. (2020), Sideris & Xinogalos (2019), Smith et al. (2019), Strawhacker & Bers (2019), Tasadduq et al. (2021), Topalli & Cagiltay (2018), Toukiloglou & Xinogalos (2022), Troiano et al. (2019), Wei et al. (2021), Wong & Yatim (2018), Xinogalos & Tryfou (2021), Yallihep & Kutlu (2020), Zatarain Cabada et al. (2020), Zhu et al. (2019), Suresh Babu & Dhakshina Moorthy (2024), Zourmpakis, Kalogiannakis & Papadakis (2023), Dehghanzadeh et al. (2024), Permana, Permatawati & Khoerudin (2023).* | Quantitative | 41 |
| *Hooshyar, Yousefi & Lim (2019), Krugel & Hubwieser (2017), Marwan, Jay Williams & Price (2019), Rakhmanita, Kusumawardhani & Anggarini (2023), Kitani (2023)* | Qualitative | 05 |
| *Pradana et al. (2023)* | Quasi-experimental | 1 |
| *Cao (2023), Ghosh & Pramanik (2023), Rodrigues et al. (2023), Shortt et al. (2023)* | Quantitative + Qualitative | 4 |
| *Zhang & Hasim (2023)* | Quantitative + Qualitative + Quasi-experimental | 1 |

**Table 13 Related studies evaluation measures.**

| Item No. | Ref. | Game | Measured approach | Result presented | Software |
|---|---|---|---|---|---|
| 1 | *Giannakoulas & Xinogalos (2018)* | Run marco game | Effectiveness and acceptance | Descriptive | SPSS |
| 2 | *Zatarain Cabada et al. (2020)* | EasyLogic | – | Descriptive and t-test, regression analysis | SEM |
| 3 | *Malliarakis, Satratzemi & Xinogalos (2017)* | CMX environment | Effectiveness | Descriptive, Mean, S. D Correlation | SPSS |
| 4 | *Papadakis & Kalogiannakis (2019)* | Dr. Scratch | Evaluation | Mean, S. D | SPSS |
| 5 | *Topalli & Cagiltay (2018)* | Dr. Scratch | Improvement | t test | SPSS |
| 6 | *Chang, Chung & Chang (2020)* | Programmer adventure land | Effectiveness | t test | SPSS |
| 7 | *Jakoš & Verber (2017)* | Aladdin and his flying carpet | Improvement | Mean, S.D, Paired-samples t test, ANOVA | SPSS, Excel |
| 8 | *Garneli & Chorianopoulos (2018)* | Dr. Scratch | Exploring | Non-parametric Wilcoxon signed-rank test (z) and non-parametric Mann–Whitney U test | SPSS |
| 9 | *Mathew, Malik & Tawafak (2019)* | PROSOLVE | Problem solving | Descriptive | SPSS |
| 10 | *Pellas & Vosinakis (2018)* | Scratch and OpenSim with the Scratch4SL palette | Effectiveness | Mean, S.D, Mann-Whitney U | SPSS |
| 11 | *Wei et al. (2021)* | Computational thinking with scratch | Effectiveness | ANCOVA | SPSS |
| 12 | *Strawhacker & Bers (2019)* | Dr. Scratch | Investigation | Non-parametric Kruskal–Wallis H, or Kruskal–Wallis | SPSS |
| 13 | *Hitchens & Tulloch (2018)* | A software | – | – | – |

| Item No. | Ref. | Game | Measured approach | Result presented | Software |
|---|---|---|---|---|---|
| 14 | *Syaifudin, Funabiki & Kuribayashi (2019)* | Android programming learning assistance system | Test-driven development method | – | – |
| 15 | *Marwan, Jay Williams & Price (2019)* | iSnap | Evaluation | Interview | – |
| 16 | *Maskeliūnas et al. (2020)* | – | Effectiveness | – | – |
| 17 | *Duffany (2017)* | – | – | – | – |
| 18 | *Seraj, Autexier & Janssen (2018)* | BEESM | – | – | – |
| 19 | *Figueiredo & García-Peñalvo (2018)* | – | – | – | – |
| 20 | *Krugel & Hubwieser (2017)* | MOOC called LOOP | Computational thinking | Textual feedback | – |
| 21 | *Skalka & Drlík (2018)* | – | – | – | – |
| 22 | *Nadolny et al. (2017)* | – | – | – | – |
| 23 | *Hooshyar, Yousefi & Lim (2019)* | Online game-based bayesian intelligent tutoring system | Evaluation | Interview | – |
| 24 | *Von Hausswolff (2017)* | – | – | – | – |
| 25 | *Drosos, Guo & Parnin (2017)* | HappyFace | Identification | – | – |
| 26 | *Bernik, Radošević & Bubaš (2017)* | – | – | – | – |
| 27 | *Troiano et al. (2019)* | Dr. Scratch | Evaluation | Descriptive, cluster analysis, and data visualization | – |
| 28 | *Malik et al. (2019)* | PROBSOL | Problem solving | – | – |
| 29 | *Devine et al. (2019)* | MS MakeCode and CODAL | Evaluation | – | – |
| 30 | *Yallihep & Kutlu (2020)* | Lightbot | Effectiveness | Descriptive, t test | SPSS |
| 31 | *Piedade et al. (2020)* | – | Computational thinking | – | – |
| 32 | *Luik et al. (2019)* | – | – | – | – |
| 33 | *Luxton-Reilly et al. (2019)* | – | Check pass rate | – | – |
| 34 | *Martins, de Almeida Souza Concilio & de Paiva Guimarães (2018)* | – | Problem based learning | Descriptive | SPSS |
| 35 | *Smith et al. (2019)* | – | Effect | Descriptive, linear regression, correlation | SPSS |
| 36 | *Schez-Sobrino et al. (2020)* | RoboTIC | Motivation | Descriptive | SPSS |
| 37 | *Ivanović et al. (2017)* | LMS | Effectiveness | Kruskal-Wallis ANOVA, Mann-Whitney U Kolmogorov-Smirnov test | SPSS |
| 38 | *Hellings & Haelermans (2020)* | – | Effect | Descriptive, regression | SPSS |
| 39 | *Marwan, Jay Williams & Price (2019)* | iSnap | Impact | Descriptive | SPSS |
| 40 | *Laporte & Zaman (2018)* | – | – | – | – |
| 41 | *Kumar & Sharma (2018)* | ProLounge | Achievement | Descriptive results | Excel |
| 42 | *de Pontes, Guerrero & de Figueiredo (2019)* | – | – | – | – |
| 43 | *Paiva, Leal & Queirós (2020)* | – | Impact | Descriptive, Mann-Whitney U one-sided tests | SPSS |
| 44 | *Wong & Yatim (2018)* | Odyssey of Phoenix | Learning | Descriptive, paired sample T-test, ANOVA | SPSS |

(Continued)

| Item No. | Ref. | Game | Measured approach | Result presented | Software |
|---|---|---|---|---|---|
| 45 | *Gulec et al. (2019)* | CENGO | Achievement | Descriptive | – |
| 46 | *Tasadduq et al. (2021)* | CYourWay | Effect | Descriptive, independent sample t test, Mann-Whitney u test | SPSS |
| 47 | *Abbasi et al. (2021)* | POOsg | Performance, motivation | Descriptive, Paired t-test | SPSS |
| 48 | *Zhu et al. (2019)* | Parallel | Effectiveness | Descriptive | SPSS |
| 49 | *Sideris & Xinogalos (2019)* | PY-RATE ADVENTURES | Learning | Descriptive | SPSS |
| 50 | *Montes et al. (2021)* | DFD-C | Effectiveness | Descriptive | SPSS |
| 51 | *Xinogalos & Tryfou (2021)* | Game of code: lost in Javaland | Motivation | Descriptive | SPSS |
| 52 | *Toukiloglou & Xinogalos (2022)* | Dungeon class | Effectiveness | Frequency, Kruskal–Wallis test | SPSS |
| 53 | *Daungcharone, Panjaburee & Thongkoo (2017)* | CPGame | Effectiveness | Mean, SD, MANOVA | SPSS |
| 54 | *Carreño-León, Rodríguez-Álvarez & Sandoval-Bringas (2019)* | – | Effectiveness | Frequencies | SPSS |
| 55 | *Jemmali et al. (2019)* | May's journey 3D puzzle game | Learning | – | – |
| 56 | *Khaleel, Ashaari & Wook (2019)* | Gami-PL | Effectiveness, motivation | Mean, SD, t test | SPSS |
| 57 | *Moreno & Pineda (2018)* | Gamification activities | learning | Mean, SD, Skewness, Kurtis | SPSS |
| 58 | *Pankiewicz (2020)* | – | Impact | U Mann-Whitney test | SPSS |
| 59 | *Queirós (2019)* | PROud framework | – | – | – |
| 60 | *Marín et al. (2019)* | UDPiler | Investigation | Descriptive | SPSS |
| 61 | *Huseinović (2024)* | Gamification activities | Impact | Descriptive | SPSS |
| 62 | *Shortt et al. (2023)* | Duolingo application | Investigation | Descriptive | SPSS |
| 63 | *Pradana et al. (2023)* | HSS gamification platform | Investigation | Descriptive | |

nominal studies used Microsoft Excel. The methodology and tools used by the selected studies are also presented in Table 13. Portions of this text were previously published as part of a preprint (*Ishaq & Alvi, 2023*).

**Evaluation measures:**

In this section, the evaluation measures terminologies are described from the selected studies:

*Descriptive:* Descriptive statistics are short informative coefficients that describe a specific data collection, which might represent the full population or a subset of a population (*Hayes, 2022*). The descriptive results presented by *Abbasi et al. (2021)*, *Giannakoulas & Xinogalos (2018)*, *Gulec et al. (2019)*, *Malliarakis, Satratzemi & Xinogalos (2017)*, *Marín et al. (2019)*, *Martins, de Almeida Souza Concilio & de Paiva Guimarães (2018)*, *Marwan, Jay Williams & Price (2019)*, *Mathew, Malik & Tawafak (2019)*, *Montes et al. (2021)*, *Paiva, Leal & Queirós (2020)*, *Schez-Sobrino et al. (2020)*, *Sideris & Xinogalos (2019)*, *Smith et al. (2019)*, *Tasadduq et al. (2021)*, *Troiano et al. (2019)*, *Wong & Yatim (2018)*, *Xinogalos & Tryfou (2021)*, *Yallihep & Kutlu (2020)*, *Zatarain Cabada et al. (2020)*, and *Zhu et al. (2019)*

*Frequencies:* A frequency distribution is a visual or tabular display that shows the number of occurrences over a specific period of time (*Young, 2020*) calculated by *Carreño-León, Rodríguez-Álvarez & Sandoval-Bringas (2019)* and *Toukiloglou & Xinogalos (2022)*.

*Mean:* Mean which is the average of the data set (adding all the numbers then dividing by its total point) (*Wei, 2020*) was calculated by *Daungcharone, Panjaburee & Thongkoo (2017)*, *Jakoš & Verber (2017)*, *Khaleel, Ashaari & Wook (2019)*, *Moreno & Pineda (2018)*, *Papadakis & Kalogiannakis (2019)*, and *Pellas & Vosinakis (2018)*.

*Standard Deviation (SD):* SD is the square root of the variance, which measures how to spread out a set of numbers is compared to its mean (*Hargrave, 2020*). It was calculated by *Daungcharone, Panjaburee & Thongkoo (2017)*, *Jakoš & Verber (2017)*, *Khaleel, Ashaari & Wook (2019)*, *Malliarakis, Satratzemi & Xinogalos (2017)*, *Moreno & Pineda (2018)*, *Papadakis & Kalogiannakis (2019)*, and *Pellas & Vosinakis (2018)*.

*t-test:* The independent t-test compares two collections of data, each of which is centered on a constant value, to determine whether or not there is statistical significance between them (*e.g.*, interval or ratio) (*Statistics Solutions, 2013b*) was calculated by *Abbasi et al. (2021)*, *Chang, Chung & Chang (2020)*, *Khaleel, Ashaari & Wook (2019)*, *Tasadduq et al. (2021)*, *Topalli & Cagiltay (2018)*, *Wong & Yatim (2018)*, *Yallihep & Kutlu (2020)*, and *Zatarain Cabada et al. (2020)*

*Analysis of variance:* The statistical technique known as analysis of variance (ANOVA) is used to compare multiple groups using a dependent variable that has two or more discrete categories (*Statistics Solutions, 2013a*), which were calculated by *Jakoš & Verber (2017)*, and *Wong & Yatim (2018)*.

*Analysis of covariance:* A continuous variable is added to the variables of interest in an analysis of covariance (ANCOVA) (*i.e.*, the dependent and independent variable) as means for control (*Statistics Solutions, 2013c*) calculated by *Wei et al. (2021)*.

*Multivariate analysis of variance:* The goal of multivariate analysis of variance (MANOVA), which is similar to ANOVA, is to examine differences between groups by using two or more dependent variables as opposed to one metric dependent variable (*Statistics Solutions, 2013d*) calculated by *Daungcharone, Panjaburee & Thongkoo (2017)*.

*Wilcoxon signed-rank test (z):* Two paired groups can be compared using the nonparametric Wilcoxon test, which can be either the rank sum test or the signed-rank test. The tests effectively compute the difference between groups of pairings and examine this difference to see if it is statistically significant (*Hayes, 2021*) calculated by *Garneli & Chorianopoulos (2018)*.

*Mann–Whitney U test:* When the dependent variable is ordinal or continuous but not normally distributed, the Mann-Whitney U test is used to examine the differences between two groups (*Statistics Solutions, 2021*) calculated by *Garneli & Chorianopoulos (2018)*, *Paiva, Leal & Queirós (2020)*, *Pankiewicz (2020)*, *Pellas & Vosinakis (2018)*, and *Tasadduq et al. (2021)*.

*Kruskal–Wallis:* The medians of three or more independent groups are compared using the Kruskal-Wallis test to evaluate whether or not there is a statistically significant difference (*Zach, 2022*) calculated by *Strawhacker & Bers (2019)*, and *Toukiloglou & Xinogalos (2022)*.

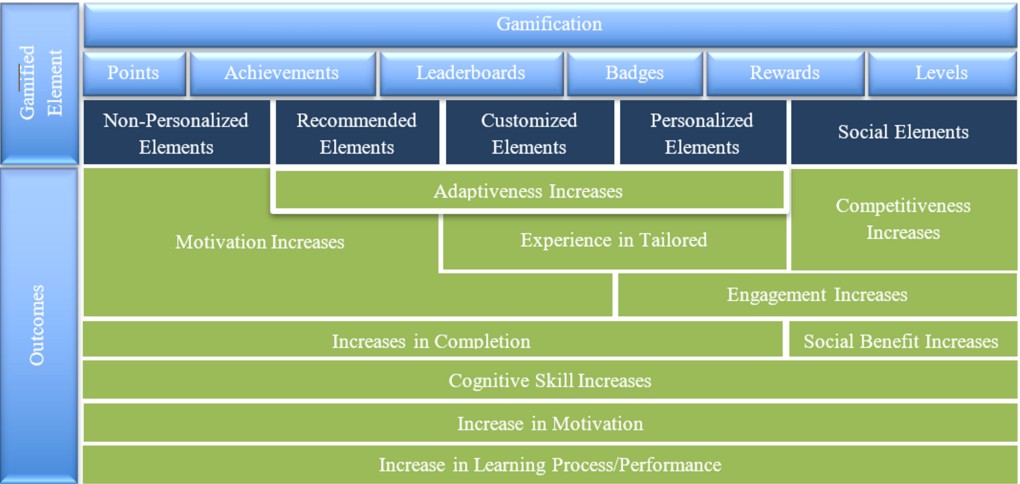

**Figure 3 Diagram of gamification elements and their outcomes.**

*Linear regression:* The purpose of a linear regression analysis is to determine if one or more predictor variables can account for the presence or absence of a certain dependent (criterion) variable (*Statistics Solutions, 2013e*) calculated by *Hellings & Haelermans (2020)*, *Smith et al. (2019)*, and *Zatarain Cabada et al. (2020)*.

*Correlation:* Correlation is a statistical term that reflects how much two or more variables change in relation to each other (*Wigmore, 2020*), which was calculated by *Malliarakis, Satratzemi & Xinogalos (2017)* and *Smith et al. (2019)*.

Overall, this systematic literature review provides valuable insights into the trends, best practices, and impacts of personalized gamified programming education on students' cognition. Most of the studies used various tools to evaluate programming language learning in their respective areas, including questionnaires, interviews, and observation methods. The findings of this research question showed that most of the studies used questionnaire surveys and SPSS tools for data analysis.

## DISCUSSION AND FUTURE IMPLICATIONS

The consensus on the current state of the plethora of gamification in education research is that gamification consistently improves motivation and performance, as shown in Fig. 3. Portions of this text were previously published as part of a preprint (*Ishaq & Alvi, 2023*).

The results of this systematic literature review have shed light on the importance of gamification, personalization, and cognition in programming language education. The findings suggest that gamification techniques can enhance programming education engagement, motivation, and learning outcomes. Personalization of gamified programming education has also been identified as a key factor in improving student performance and satisfaction. Moreover, the results have shown that gamification can be tailored to different cognitive levels of Bloom's taxonomy to promote higher-order

thinking skills. Personalized gamification frameworks can also help students learn at their own pace and provide a more enjoyable and rewarding learning experience. Furthermore, programming language education can be enriched using various gamification techniques, such as game elements, game design principles, and game-based learning approaches. The results also suggest that different programming languages require different gamification strategies to be effective. In conclusion, the findings of this systematic literature review indicate that gamification and personalization are promising strategies for enhancing programming language education. The results also highlight the importance of considering cognitive factors when designing gamified programming education. Further research is needed to explore the effectiveness of different gamification strategies in various programming languages and to evaluate the impact of personalized gamified programming education on student learning outcomes.

## FINDINGS, CHALLENGES, AND RECOMMENDATIONS

The systematic literature review revealed several key findings regarding personalized gamified programming education. First, it was found that personalized gamification strategies can improve student engagement and motivation in programming education. Second, personalized gamification can enhance students' problem-solving and cognitive abilities. Third, gamified applications' design and customization can significantly impact personalized gamification strategies' effectiveness. Fourth, there is a need for more empirical studies to validate the effectiveness of personalized gamification strategies in programming education. Finally, the review identified a lack of consensus on the evaluation criteria and metrics for assessing the quality of personalized gamification applications in programming education. Several challenges were identified during the systematic literature review. One of the primary challenges is the limited availability of high-quality research on personalized gamified programming education. Additionally, the lack of standardization in designing and evaluating gamified applications makes comparing the effectiveness of different personalized gamification strategies difficult. Another challenge is the need for skilled instructors who can effectively design and implement personalized gamification strategies in programming education. Based on the findings and challenges identified in this systematic literature review, the following recommendations are made:

*Empirical studies:*

Personalized gamification strategies have shown promise in enhancing engagement and learning outcomes in programming education. However, there is a significant gap in empirical evidence supporting their effectiveness. To bridge this gap, more rigorous and comprehensive empirical studies are needed that should aim to:

- Assess the long-term impact of personalized gamification on student motivation and learning outcomes.
- Explore how different demographic factors, such as age, gender, and prior programming experience, influence the effectiveness of personalized gamification.

- Compare personalized gamification strategies with traditional teaching methods and other educational technologies to determine their relative effectiveness.

*Design and evaluation standard:*

The field of personalized gamification in programming education is still in its nascent stages, leading to a lack of standardization in design and evaluation. Establishing standardized frameworks and metrics is crucial for the following reasons:

- Standardized design and evaluation methods will enable researchers to compare the effectiveness of different personalized gamification strategies more easily. This comparability will facilitate the identification of best practices and the most impactful design elements.
- Standardization will enhance the replicability of studies, allowing other researchers to validate findings and build on existing work.
- With standardized evaluation criteria, ensuring the quality and rigor of research in this area will be easier.

**Tools and resources for instructors:**

Instructors need access to practical tools and resources for personalized gamification strategies to be widely adopted in programming education. These tools and resources should:

- Provide templates and guidelines for designing personalized gamification elements, such as adaptive quizzes, progress-tracking dashboards, and personalized feedback mechanisms.
- Offer user-friendly platforms and software that integrate seamlessly with existing Learning Management Systems (LMS). These platforms should allow instructors to implement and customize gamification elements easily.
- Include training programs and workshops to help instructors understand the principles of personalized gamification and develop the necessary skills to design and implement these strategies effectively.

**Identifying effective strategies:**

Programming education encompasses a wide range of cognitive levels, from beginner to advanced, and a variety of programming languages, each with its unique challenges. Future research should aim to:

- Identify which personalized gamification strategies are most effective for different cognitive levels. For example, beginners might benefit more from gamification elements that simplify complex concepts, while advanced students might respond better to challenges that encourage deeper problem-solving.
- Determine how personalized gamification strategies can be tailored to specific programming languages. Some languages may lend themselves more readily to certain types of gamification due to their syntax, complexity, or application areas.

## CONCLUSIONS

In conclusion, this systematic literature review highlights the importance of gamification, personalization, cognition, and programming education in augmenting students' educational outcomes. The review delineates various trends and optimal methodologies for implementing personalized gamification frameworks within programming education, emphasizing their role in enhancing students' cognitive proficiencies. Nonetheless, the review elucidates certain challenges inherent in gamification and personalization in programming education, notably the necessity for tailored tools and software applications tailored to specific programming languages and concepts. Furthermore, it underscores gaps in extant scholarship, including the paucity of research on the enduring effects of personalized gamified programming education and the dearth of investigations into the efficacy of gamification across diverse programming languages. To surmount these challenges and bridge existing gaps in programming education, we advocate for concerted efforts among researchers and educators to devise bespoke gamification strategies and software tools attuned to the unique exigencies of programming learners. Additionally, we underscore the imperative for expanded research endeavors to elucidate the enduring impacts of gamified programming education and evaluate the efficacy of gamification across a spectrum of programming languages. Moreover, we underscore the utility of game-based learning beyond programming education, citing its efficacy in language acquisition, healthcare, business, and marketing, wherein it facilitates immersive simulations and experiential learning. Finally, we advocate for incorporating cognitive considerations into the development of tailored gamification frameworks within programming education, thereby fostering more effective and targeted educational interventions.

## ACKNOWLEDGEMENTS

I would like to thank all the authors who contributed to conducting this detailed study and provided valuable feedback. Moreover, the initial draft of this study has been published in arXiv under the identification number 2309.12362.

### Funding

The research was supported by the grant: FRGS/1/2019/ICT01/UKM/01/1. The funders had no role in study design, data collection and analysis, decision to publish, or preparation of the manuscript.

### Grant Disclosures

The following grant information was disclosed by the authors:
FRGS/1/2019/ICT01/UKM/01/1.

### Competing Interests

The authors declare that they have no competing interests.

## Author Contributions

- Kashif Ishaq conceived and designed the experiments, performed the experiments, prepared figures and/or tables, and approved the final draft.
- Atif Alvi conceived and designed the experiments, analyzed the data, performed the computation work, prepared figures and/or tables, authored or reviewed drafts of the article, and approved the final draft.
- Muhammad Ikram ul Haq conceived and designed the experiments, performed the experiments, performed the computation work, authored or reviewed drafts of the article, and approved the final draft.
- Fadhilah Rosdi performed the experiments, analyzed the data, authored or reviewed drafts of the article, and approved the final draft.
- Abubakar Nazeer Choudhry conceived and designed the experiments, performed the computation work, authored or reviewed drafts of the article, and approved the final draft.
- Arslan Anjum performed the experiments, prepared figures and/or tables, and approved the final draft.
- Fawad Ali Khan analyzed the data, performed the computation work, prepared figures and/or tables, and approved the final draft.

## Data Availability

This is a literature review.

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
