# Peer review of "Level up your coding: a systematic review of personalized, cognitive, and gamified learning in programming education"

_PeerJ Computer Science, doi:10.7717/peerj-cs.2310_

## Round 0.1 · original submission · Minor Revisions

Based on the reviewers' comments, authors are recommended to carry out "Minor Revision" on this paper by addressing the issues raised by the reviewers in particular the start date of this research (why 2014 ?) and also the latest research papers should be incorporated into this review.

·

Basic reporting

The proposed paper is a typical systematic review of literature on some topics. I find the structure of the paper straightforward. It follows the standard pattern and methodology of systematic literature review, like employing multiple reviewers, clear inclusion and exclusion criteria, snowballing, datasets, and query descriptions.

I am not a native English speaker, but I did not find any issues regarding that.

Experimental design

All personalization, cognition, and gamification are diverse topics and require more detailed elaboration. The gamification is not all about badges, resource gathering, etc. Some game design elements, like narrative and storytelling, social interaction, collaboration, and others, are also important. Personalization involves adaptive learning technologies, dynamic assessments, and others. Cognition, among others, involves conceptual understanding, cognitive load, misconceptions, and metacognition. That also includes knowledge of psychology and pedagogy. The other dimensions are the learners' age, and the programming languages taught. The pupils in elementary schools have different learning profiles than the students. Furthermore, Scratch, for example, requires a mindset different from Python programming.

It would be beneficial if you could extend the text to describe the basic topics of the research. I would also suggest including a more extensive discussion about the topics listed as recommendations in Chapter 6 in the main part of the paper.

Validity of the findings

See the previous comment.

Additional comments

I acknowledge that the author has been involved intensively in the research. Nevertheless, the Web of Science is not the only relevant source of quality articles. We encourage our students to include multiple data sources. Especially in cognition, pedagogy, dealing with misconceptions, and more, some useful research studies are not included in the WoS and are usually not caught by snowballing. However, this is just a comment for future work.

Reviewer 2 ·

Basic reporting

1.- Add more references in the introducción and background because there are some affirmations that are not back up by the literature.. Eg. "Gamification involves integrating game-like elements into the learning process to enhance engagement and motivation."... No author...
2.- Section 1.2 is not clear to me. The other 2 sections are clear because your answers focus on those areas but I do not find a connection with this section. Your RQ do not look for that information either. You could just mention some of thi section as background information but not an exclusive section.

Experimental design

Congratulations on the design. Very detailed!. My only comment is to justify why the selection criteria started in 2014 and not before.

Validity of the findings

I think this part should be avoided. "The consensus on the current state of the plethora of gamification in education research is that gamification consistently improves motivation and performance". The research questions never addressed that question and the results never show anything related to the aspects that gamification improves. Thus, the table should also be deleted. This is why, section 1.2 should be changed. This article focuses on other things like cognitive levels, personalization, etc.

Additional comments

Good Job on the article.

---

## Round 0.2 · Minor Revisions

Although authors have revised the paper, but it is not clear what authors have done with comments. How these comments were implemented in the revised manuscript?

Please clarify this in the responses. Specifically, please provide a new rebuttal letter which explicitly notes the line numbers / sections of your manuscript where you have made the changes.

Reviewer 2 ·

Basic reporting

No comment

Experimental design

No comment

Validity of the findings

No comment

Additional comments

All my comments have been addressed.

---

## Round 0.3 · accepted · Accept

Authors have addressed all the comments adequately. Hence, this paper can be accepted.